# The ORP9-ORP11 dimer promotes sphingomyelin synthesis

Birol Cabukusta[1]*, Shalom Borst Pauwels[1], Jimmy JLL Akkermans[1], Niek Blomberg[2], Aat A Mulder[3], Roman I Koning[3], Martin Giera[2], Jacques Neefjes[1]

[1]Cell and Chemical Biology, Oncode Institute, Leiden University Medical Center, Leiden, Netherlands; [2]Centre for Proteomics and Metabolomics, Leiden University Medical Center, Leiden, Netherlands; [3]Electron Microscopy Facility, Cell and Chemical Biology, Leiden University Medical Center, Leiden, Netherlands

**\*For correspondence:**
b.cabukusta@lumc.nl

**Abstract** Numerous lipids are heterogeneously distributed among organelles. Most lipid trafficking between organelles is achieved by a group of lipid transfer proteins (LTPs) that carry lipids using their hydrophobic cavities. The human genome encodes many intracellular LTPs responsible for lipid trafficking and the function of many LTPs in defining cellular lipid levels and distributions is unclear. Here, we created a gene knockout library targeting 90 intracellular LTPs and performed whole-cell lipidomics analysis. This analysis confirmed known lipid disturbances and identified new ones caused by the loss of LTPs. Among these, we found major sphingolipid imbalances in ORP9 and ORP11 knockout cells, two proteins of previously unknown function in sphingolipid metabolism. ORP9 and ORP11 form a heterodimer to localize at the ER-*trans*-Golgi membrane contact sites, where the dimer exchanges phosphatidylserine (PS) for phosphatidylinositol-4-phosphate (PI(4)P) between the two organelles. Consequently, loss of either protein causes phospholipid imbalances in the Golgi apparatus that result in lowered sphingomyelin synthesis at this organelle. Overall, our LTP knockout library toolbox identifies various proteins in control of cellular lipid levels, including the ORP9-ORP11 heterodimer, which exchanges PS and PI(4)P at the ER-Golgi membrane contact site as a critical step in sphingomyelin synthesis in the Golgi apparatus.

## eLife assessment

This **valuable** manuscript systematically addresses the role of intracellular lipid transfer proteins on cellular lipid levels. It provides **convincing** evidence on the role of ORP9 and ORP11 in sphingolipid metabolism at the Golgi complex. This article will be of broad interest to cell biologists interested in lipid metabolism and membrane biology.

## Introduction

Many types of lipids are heterogeneously distributed among various organelles of the cell (***van Meer and de Kroon, 2011***). Due to the hydrophobic nature of lipids, their trafficking needs to be facilitated. While vesicular trafficking transports lipids in bulk between organelles, non-vesicular trafficking of lipids also plays a role in defining the lipid compositions of organelles. In fact, many organelles that do not receive any vesicular cargo rely on this route. Non-vesicular trafficking of lipids is achieved by a group of LTPs that carry lipids using their hydrophobic cavities to stabilize lipids in the aqueous intracellular environment (***Wong et al., 2019***). LTPs often contain targeting domains, motifs, transmembrane regions, amphipathic helices, or surface charges to define their donor and acceptor organelles (***Chiapparino et al., 2016***). Many LTPs localize at two organelles simultaneously at membrane

contact sites, where these organelles come closer to exchange information and material, including lipids (*Scorrano et al., 2019*; *Prinz et al., 2020*).

The human genome encodes over 150 LTPs; new LTPs and LTP families are being characterized every so often (*Wong et al., 2019*; *Gatta et al., 2015*; *Sandhu et al., 2018*; *Valverde et al., 2019*; *Osawa et al., 2019*; *Yeo et al., 2021*; *Kumar et al., 2018*; *Castro et al., 2022*; *Neuman et al., 2022*). About 50 of these proteins are secreted and are involved in carrying lipids, metals, lipopolysaccharides, and other small molecules in blood plasma (*Wong et al., 2019*). The remaining intracellular LTPs are mainly responsible for intracellular lipid trafficking among organelles. Since many lipid-modifying enzymes are localized to different organelles, LTPs also feed metabolic lipid fluxes, eventually defining cellular lipid levels (*Holthuis and Menon, 2014*). For example, LTP-mediated trafficking of ceramide from the endoplasmic reticulum (ER) to the *trans*-Golgi is needed for sphingomyelin synthesis (*Hanada et al., 2003*). While achieving a considerable understanding of some LTPs in the last years, the function of many LTPs in defining cellular levels and distributions remains unclear.

To study LTPs systematically, we designed and created an arrayed gene knockout library targeting 90 intracellular LTPs based on the CRISPR/Cas9 technology. Lipidomics analysis of the library-generated LTP knockout cells confirmed known and identified novel lipid disturbances emerging from loss of LTPs. These included CERT, GLTP, NPC1, and NPC2 knockout cells with altered sphingolipid levels. Furthermore, we identified major sphingolipid imbalances in ORP9 and ORP11 knockout cells, two proteins of unknown function in sphingolipid metabolism. ORP9 and ORP11 form a heterodimer to localize at the ER-*trans*-Golgi membrane contact sites. At this contact site, the ORP9-ORP11 dimer transfers PS from the ER to the Golgi and PI(4)P in the opposite direction. Consequently, loss of either protein causes phospholipid imbalances in the Golgi apparatus that result in lowered sphingomyelin synthesis capacity at this organelle. Collectively, our LTP knockout library toolbox identified various proteins controlling cellular lipid levels. Among these, we found the ORP9-ORP11 heterodimer defining phospholipid composition of the *trans*-Golgi as a critical step in sphingomyelin synthesis. These findings highlighted that phospholipid and sphingolipid gradients along the secretory pathway are linked at the ER-Golgi membrane contact sites.

## Results

### A CRISPR knockout library targeting lipid transfer proteins

To understand the function of LTPs in defining cellular levels and distributions, we created an arrayed CRISPR/Cas9 knockout library targeting LTPs. We targeted only the intracellular LTPs in the library and excluded extracellular proteins, such as LIPOCALIN and LBP-BPI-CETP families *Wong et al., 2019*. In this library, 90 wells of a 96-well plate were used for targeting LTPs and 6 for non-targeting (NT) controls (*Figure 1A*). For optimal gene disruption, we used a lentiviral gene delivery system and three guide RNAs per gene – a method also used by others to increase the frequency of gene disruption (*Wong et al., 2023*). We applied the LTP knockout library to MelJuSo, a human melanoma cell line. The efficiency of the knockout strategy was confirmed by western blotting of selected wells (*Figure 1B*).

Next, we performed a whole-cell lipidomics analysis of the LTP knockout cells. To this end, library-generated MelJuSo cells were grown in lipid-depleted serum and analyzed using a lipidomics platform detecting 17 different lipid classes thrice on average (*Figure 1A*). One of the detected lipid classes, hexosylceramide, can correspond to glucosylceramide and galactosylceramide – two isomeric lipids in mammalian cells. By metabolic chasing of a fluorescent ceramide analogue in cells silenced for glucosylceramide synthase or galactosylceramide synthase; or treated with the glucosylceramide synthase inhibitor PDMP, we validated that glucosylceramide is the major hexosylceramide in MelJuSo cells (*Figure 1—figure supplement 1*). This further substantiated the previous reports of galactosylceramide being present mainly in oligodendrocytes (*Reza et al., 2021*).

We next calculated the z-scores within each lipid class based on their relative abundance per measurement and plotted their average for LTP knockout cell lines. The analysis revealed many LTP knockout cells with lipid imbalances (*Figure 1C*, *Figure 1—figure supplement 2A*). Meanwhile, no z-score above the absolute value of 2 was observed for NT control cells (*Figure 1—figure supplement 2B*). Overall, the lipidomics analysis of the LTP knockout library uncovered numerous candidate regulators of cellular lipid levels.

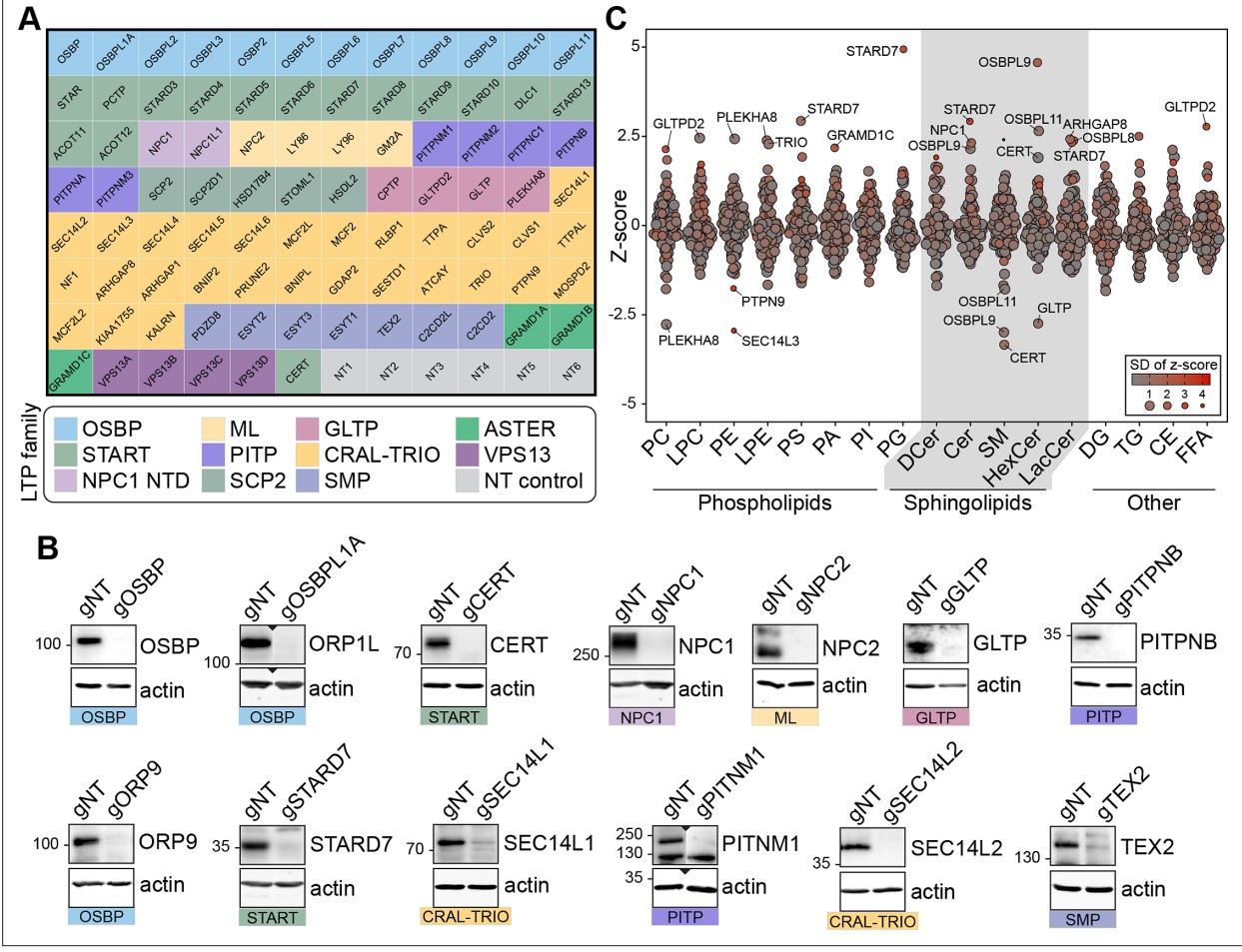

**Figure 1.** Lipidomics analysis of a lipid transfer protein (LTP) knockout library revealed known and candidate controllers of cellular lipid levels. (**A**) Overview of the arrayed gene knockout library targeting LTPs. NT: non-targeting. (**B**) Western blotting of the selected wells showing the efficiency of the LTP knockout library on MelJuSo cells. gNT: Non-targeting, control cells. (**C**) Summary plot of the lipidomics analysis: Z-scores calculated for 17 lipid classes are plotted. Each data point represents the mean Z-score of an LTP knockout cell line; size and color of data points represent the standard deviation of Z-scores for three experiments on average. PC: phosphatidylcholine, LPC: lyso-phosphatidylcholine, PE: phosphatidylethanolamine, LPE: lyso-phosphatidylethanolamine, PS: phosphatidylserine, PI: phosphatidylinositol, PA: phosphatidic acid, PG: phosphatidylglycerol, DCer: dihydroxyceramide, Cer: ceramide, SM: sphingomyelin, HexCer: hexosylceramide, LacCer: lactosylceramide, DG: diacylglycerol, TG: triacylglycerol, CE: cholesterol ester, FFA: free fatty acids.

The online version of this article includes the following source data and figure supplement(s) for figure 1:

**Source data 1.** Raw unedited gels for *Figure 1*.

**Source data 2.** Uncropped and labeled gels for *Figure 1*.

**Figure supplement 1.** Glucosylceramide is the main hexosylceramide in MelJuSo cells.

**Figure supplement 1—source data 1.** Raw unedited gels for *Figure 1—figure supplement 1*.

**Figure supplement 1—source data 2.** Uncropped and labeled gels for *Figure 1—figure supplement 1*.

**Figure supplement 2.** Overview of lipidomics analysis performed for lipid transfer protein (LTP) knockout library.

## NPC1 and NPC2 knockout cells accumulate sphingomyelin in lysosomes and the plasma membrane

Lipidomics analysis of the LTP knockout library revealed various anticipated imbalances. Among these candidate regulators of cellular lipid levels, those related to sphingolipid metabolism are notable. This is likely due to many enzymatic steps of sphingolipid metabolism being more linear and lacking the elasticity of phospholipids (*van Meer and de Kroon, 2011*; *Holthuis and Menon, 2014*). In the analysis, we identified GLTP knockout cells with lowered glucosylceramide levels (*Figure 2A*,

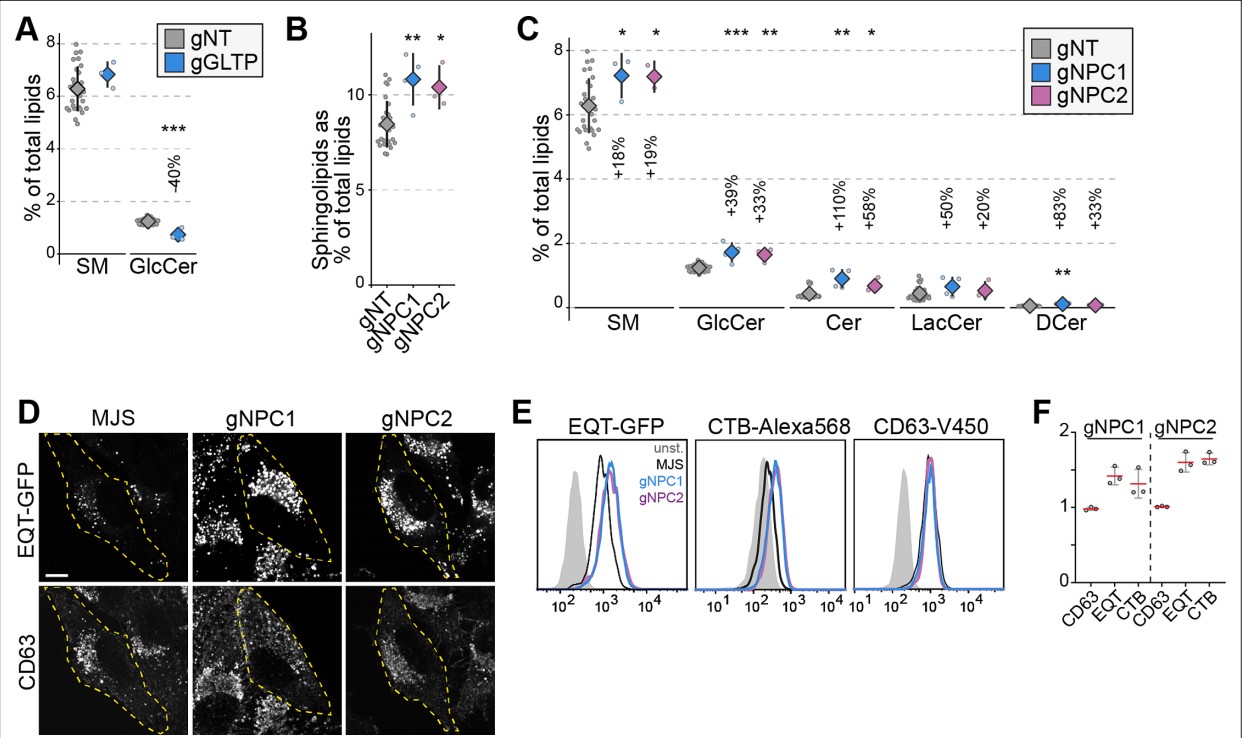

**Figure 2.** NPC1 and NPC2 knockout cells accumulate sphingomyelin in lysosomes. (**A**) GLTP knockouts show decreased glucosylceramide levels compared to control cells. GlcCer: glucosylceramide (**B**) NPC1 and NPC2 knockout cells have increased sphingolipids levels compared to control cells. (**C**) Increased sphingolipid levels in NPC1 and NPC2 knockouts are represented in all sphingolipid classes. (**D**) Immunofluorescence images of NPC1 and NPC2 knockout cells permeabilized and stained with recombinant GFP-EQT show the accumulation of sphingomyelin in the lysosomes compared to the parental MelJuSo (MJS) cells. Pixel intensities of the images were adjusted evenly. Scale bar 10 μm. (**E**) Flow cytometry analysis of cells stained with EQT-GFP, CTB-Alexa568, and CD63-V450 on the cell surface. Analysis demonstrates the accumulation of sphingomyelin (EQT-GFP) and the glycosphingolipid GM1 (CTB-Alexa568) on the cell surface without affecting the surface protein levels. (**F**) Mean fluorescence intensities for each staining normalized to control cells from three experiments. Approximately 2000 cells were analyzed per experiment. Red lines and diamonds correspond to mean mean, and the error bars denote in all figures.

The online version of this article includes the following source data and figure supplement(s) for figure 2:

**Figure supplement 1.** GLTP, NPC1, and NPC2 knockout cells do not display major phospholipid imbalances.

**Figure supplement 1—source data 1.** Raw unedited gels for *Figure 2—figure supplement 1*.

**Figure supplement 1—source data 2.** Uncropped and labeled gels for *Figure 2—figure supplement 1*.

*Figure 2—figure supplement 1A–C*). Loss of GLTP is recently reported to block ER-to-Golgi antero-grade vesicular trafficking that results in lowered glucosylceramide synthesis (*Nurmi et al., 2023*).

In addition, we observed NPC1 and NPC2 knockout cells with elevated sphingolipid levels (*Figure 1B*). Mutations in *NPC1* and *NPC2* genes are the cause of Niemann-Pick disease, type C that leads to lysosomal accumulation of cholesterol (*Platt, 2014*). Cells lacking NPC1 or NPC2 function also accumulate sphingolipids (*Newton et al., 2018*). Our observation of elevated sphingolipid levels in NPC1 and NPC2 knockout cells was present in all sphingolipid classes (*Figure 2B and C*). Meanwhile, phospholipid levels were not altered dramatically in these cells (*Figure 2—figure supplement 1D*). To confirm whether this increase is associated with lysosomal accumulation of sphingolipids, we used a sphingomyelin-binding biosensor based on the equinatoxin secreted by the beadlet anemone *Actinia equina* (*Figure 2—figure supplement 1E*; *Sokoya et al., 2022*; *Deng et al., 2016*). Immuno-fluorescence staining of NPC1 and NPC2 knockout cells using the biosensor showed lysosomal accu-mulation of sphingomyelin (*Figure 2D*). Flow cytometry analysis using the sphingomyelin biosensor and the GM1 glycosphingolipid biosensor, the B subunit of cholera toxin, demonstrated that sphingo-lipid accumulation is also reflected on the cell surface of NPC1 and NPC2 knockouts (*Figure 2E and F*). Meanwhile, the surface protein levels were not altered as detected by staining for the tetraspanin protein CD63. Collectively, our results corroborate the sphingolipid accumulation in NPC1 and NPC2

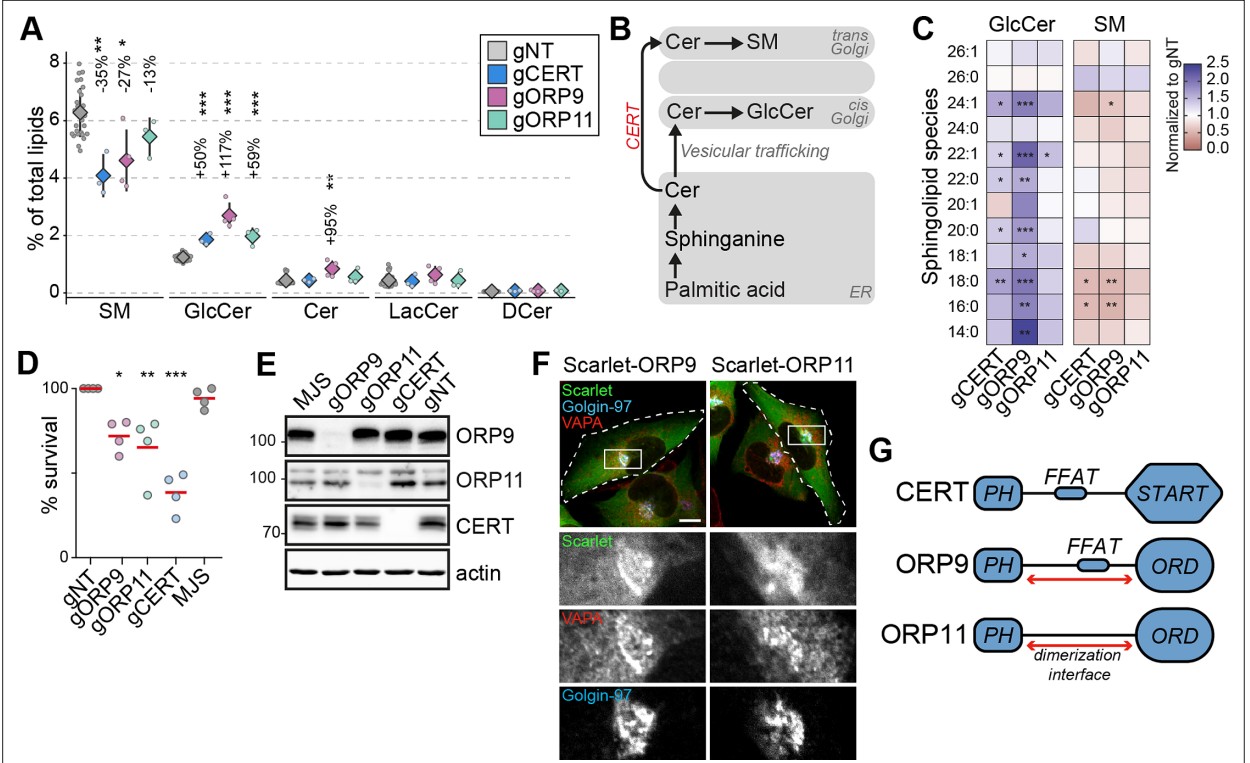

**Figure 3.** CERT, ORP9, and ORP11 knockout cells demonstrate reduced sphingomyelin levels. (**A**) CERT, ORP9, and ORP11 knockouts demonstrate decreased sphingomyelin and increased glucosylceramide levels. Diamonds and error bars denote mean and standard deviations, respectively. (**B**) Schematic representation of de novo sphingomyelin synthesis pathway of mammalian cells. (**C**) Decreased sphingomyelin and increased glucosylceramide levels observed in knockout cells are present in various sphingolipid subspecies. (**D**) Similar to CERT knockouts, ORP9 and ORP11 knockouts are sensitive to methyl-β-cyclodextrin treatment. Red lines correspond to mean values from four experiments. (**E**) Western blot of knockout cells showing that the loss of ORP9 or ORP11 does not affect CERT protein levels. (**F**) ORP9 and ORP11 localize at the Golgi apparatus. Scale bar 10 μm. (**G**) Domain architecture of CERT, ORP9, and ORP11. All proteins contain PH domains for Golgi localization. CERT and ORP9, but not ORP11, contain a FFAT motif for interacting with VAP proteins.

The online version of this article includes the following source data and figure supplement(s) for figure 3:

**Source data 1.** Raw unedited gels for *Figure 3*.

**Source data 2.** Uncropped and labeled gels for *Figure 3*.

**Figure supplement 1.** Sphingolipid imbalances in CERT, ORP9, and ORP11 knockouts are represented in multiple acyl chain species.

**Figure supplement 2.** CERT, ORP9, and ORP11 knockouts do not display defects in Golgi morphology.

deficient cells and illustrate that the LTP knockout library is a viable tool for studying LTPs and their role in regulating cellular lipid levels.

## CERT, ORP9, and ORP11 knockout cells demonstrate reduced sphingomyelin levels

Sphingomyelin is the major sphingolipid in mammalian cells, and it is mainly synthesized in the *trans*-Golgi from ceramide delivered from the ER. The non-vesicular trafficking of ceramide from the ER to the *trans*-Golgi is mediated by the ceramide transfer protein CERT (*Hanada et al., 2003*; *Kumagai and Hanada, 2019*; *Figure 3B*). Our analysis identified CERT knockouts with the lowest sphingomyelin levels (*Figures 1C and 3A*). These cells also demonstrate increased glucosylceramide levels, likely as a compensation mechanism (*Figure 3A*). Furthermore, we found decreased sphingomyelin and increased glucosylceramide levels in ORP9 and ORP11 knockout cells – two proteins of unknown role in sphingolipid metabolism (*Figures 1C and 3A*). The sphingolipid imbalances in all three knockout cells –CERT, ORP9, and ORP11– were also represented in many sphingolipid subspecies (*Figure 3C*, *Figure 3—figure supplement 1B*). Lowered sphingomyelin levels of the three knockout cells were further confirmed by their sensitivity towards methyl-β-cyclodextrin treatment (*Figure 3D*). Meanwhile,

none of the three knockout cells demonstrated phospholipid imbalances to a compelling degree (*Figure 3—figure supplement 1A*).

The observation that ORP9 and ORP11 knockout cells mimic CERT knockouts implied a role for ORP9 and ORP11 in de novo sphingomyelin synthesis, whereas loss of neither ORP9 nor ORP11 reduced CERT protein levels (*Figure 3E*). ORP9 and ORP11 each contain a PH domain mediating localization at the *trans*-Golgi, the site of de novo sphingomyelin synthesis (*Figure 3F and G*). As loss of either ORP9 or ORP11 did not alter the Golgi ultrastructure as detected by light or electron microscopy (*Figure 3—figure supplement 2*), it further suggested that ORP9 and ORP11 affect sphingomyelin levels by a previously unknown mechanism. Next, we investigated the possible mechanism by which ORP9 and ORP11 regulate cellular sphingolipid levels.

## Dimerization is critical for the ER-Golgi contact site localization of ORP9 and ORP11

ORP9 and ORP11 interact with each other via their regions between their PH and ORD domains (*Zhou et al., 2010*). As no other domain is found in this region, we speculated some secondary structures must facilitate the dimerization. AlphaFold and PCOILS coiled coils prediction tool suggested that ORP9 and ORP11 contain two alpha helices each in this region – hereafter referred to as '*coils*' (*Figure 4A*, *Figure 4—figure supplement 1A,B*). AlphaFold-Multimer suggested that the coils of ORP9 and ORP11 interact with each other (*Figure 4B*, *Figure 4—figure supplement 1C–D*). Identical analysis of the full-length proteins predicted that the coils drive the dimerization of the two proteins (*Figure 4C and D*, *Figure 4—figure supplement 1E*). This was tested by targeting the coils of ORP9 to the mitochondrial outer membrane using the *N*-terminal sequence of TOM70. In cells expressing mitochondria-targeted ORP9 coils, the coils of ORP11 are located in this organelle (*Figure 4C and D*). The absence of either coil prevented the colocalization, suggesting that the coils of ORP9 and ORP11 are sufficient for their dimerization. This finding was validated by co-immunoprecipitations (*Figure 4E*).

ORP9, but not ORP11, contains a FFAT, two phenylalanines in an acidic tract, a motif that drives ER localization by interacting with the ER-resident VAPA and VAPB (*Neefjes and Cabukusta, 2021*). Despite not containing a FFAT motif, ORP11 could interact with VAPA and VAPB by its dimerization with ORP9. We tested this by co-immunoprecipitations, where ORP11 could be co-isolated with VAPA from cells expressing VAPA, ORP9, and ORP11 (*Figure 4J*). This interaction was diminished significantly when ORP9 was not co-expressed. Furthermore, a VAPA mutant unable to interact with FFAT motifs failed to interact with either ORP9 or ORP11, showing that ORP11 interacts with VAPA indirectly via the ORP9-FFAT motif.

We and others previously showed that in addition to VAPA and VAPB, the human proteome contains three other proteins interacting with FFAT and related short linear motifs: MOSPD1, MOSPD2, and MOSPD3 (*Di Mattia et al., 2018*; *Cabukusta et al., 2020*). In our earlier efforts to identify interaction partners of VAPA, VAPB, and other motif-binding proteins, we performed BioID, proximity biotinylation followed by proteomics, and identification of membrane contact sites proteins (*Cabukusta et al., 2020*). In this analysis, ORP9 and ORP11 were found in proximity with VAPA and VAPB but not with other motif-binding proteins (*Figure 4—figure supplement 1F*). This further confirmed that despite lacking a FFAT motif, ORP11 together with ORP9 is part of VAPA- and VAPB-mediated membrane contact sites.

The Golgi localization of ORP9 and ORP11 is mediated by their PH domain interacting with phosphatidylinositol phosphates (*Zhou et al., 2010*). We observed that the PH domains of ORP9 and ORP11 localized only partially to the Golgi and demonstrated strong cytoplasmic localization, unlike the PH domains of OSBP and CERT that show exclusive localization to this organelle (*Levine and Munro, 1998*; *Levine and Munro, 2002*; *Figure 4H*; *Figure 6—figure supplement 1I*). We hypothesized that the dimerization of ORP9 and ORP11 is a mechanism to increase their avidity towards *trans*-Golgi membranes. To test this, we created a chimera containing ORP9- and ORP11-PH domains that localized to the Golgi more efficiently than the individual PH domains (*Figure 4H,I*). This also implied that loss of either protein should reduce the other's localization to the *trans*-Golgi. Immunofluorescence staining of ORP9 and ORP11 knockout cells for the same proteins confirmed that both ORP9 and ORP11 are required for the Golgi localization of the other protein (*Figure 5*). Meanwhile, loss of either protein did not influence the levels of the other (*Figure 3E*). Our results collectively show that

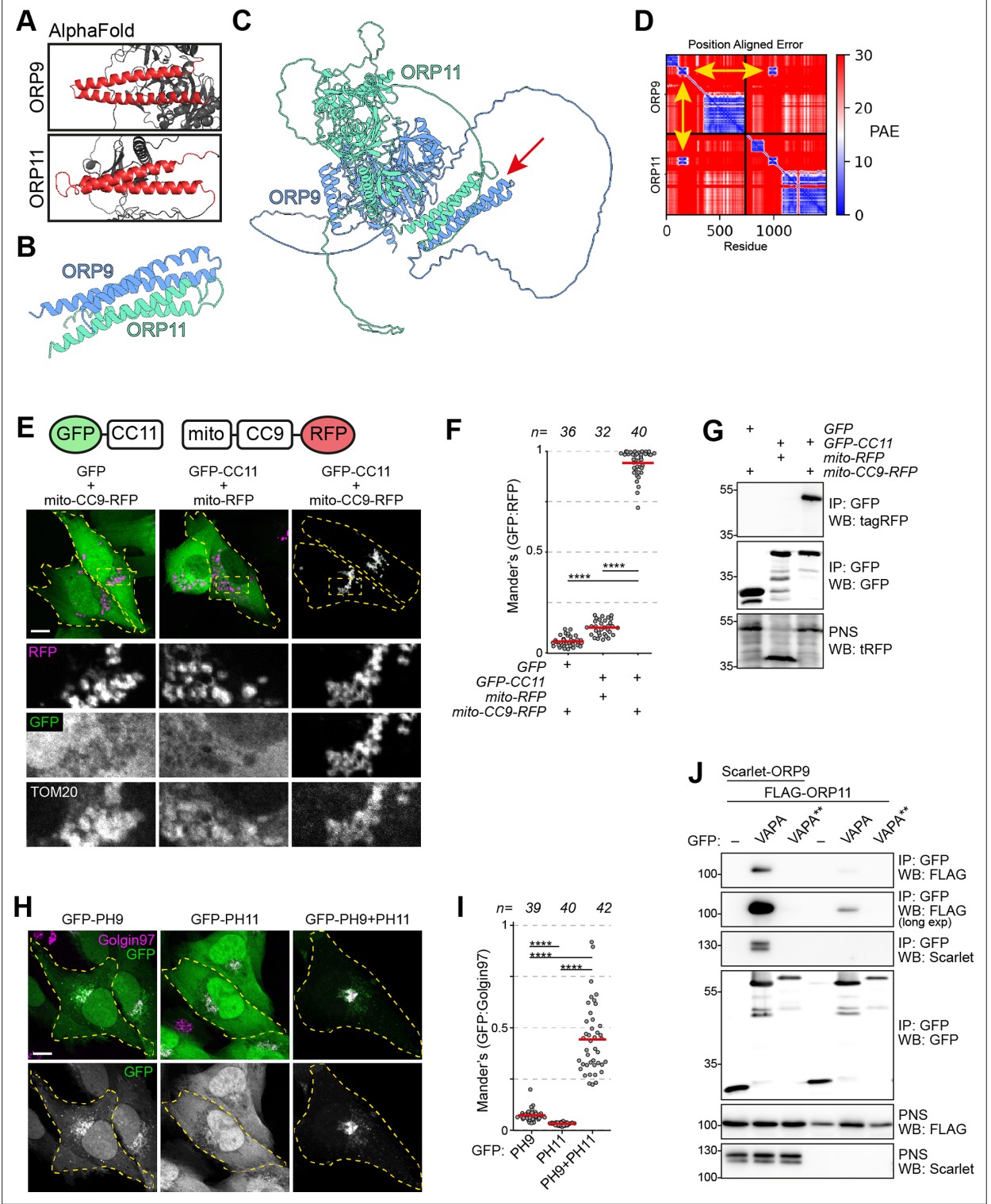

**Figure 4.** ORP9 dimerization is critical for ER localization of ORP11. (**A**) AlphaFold revealing coiled coils in ORP9 and ORP11. (**B**) AlphaFold-Multimer shows that coiled coils of ORP9 and ORP11 interacting with each other. (**C**) ColabFold protein complex prediction for full length ORP9 and ORP11 shows the dimerization via coiled coils (arrow). (**D**) Position aligned error for panel C. Note that the interaction of coils has a lower score (yellow arrows). (**E**) Coiled coils of ORP9 and ORP11 is sufficient to define their localization as ORP11 coiled coils colocalizes with the mitochondria-targeted ORP9 coiled coils at mitochondria. Scale bar: 10 µm. (**F**) Colocalization analysis of immunofluorescent images corresponding to panel E. Red lines correspond to mean values from three experiments; n is the number of analyzed cells. (**G**) Co-immunoprecipitation confirming that the coils of ORP9 and ORP11 interact with each other. (**H**) Immunofluorescence images of PH domain localizations. Compared to individual PH domains of ORP9 and ORP11, ORP9-

*Figure 4 continued on next page*

*Figure 4 continued*

ORP11 chimera demonstrates better affinity towards the Golgi. Scale bar: 10 μm. (**I**) Colocalization analysis of immunofluorescent images corresponding to panel H. Red lines correspond to mean values from 3 experiments; n is the number of analyzed cells. (**J**) Co-immunoprecipitation analysis of VAPA, ORP9, and ORP11 from over-expressing cells. Despite lacking a FFAT motif, ORP11 interacts with the ER-resident VAPA protein. This interaction is facilitated by the FFAT motif of ORP9 as a VAPA mutant unable to interact with FFAT motifs (VAPA**) was unable to co-precipitate ORP9 or ORP11.

The online version of this article includes the following source data and figure supplement(s) for figure 4:

**Source data 1.** Raw unedited gels for *Figure 4*.

**Source data 2.** Uncropped and labeled gels for *Figure 4*.

**Figure supplement 1.** ORP9 and ORP11 interact with each other via their coiled coils.

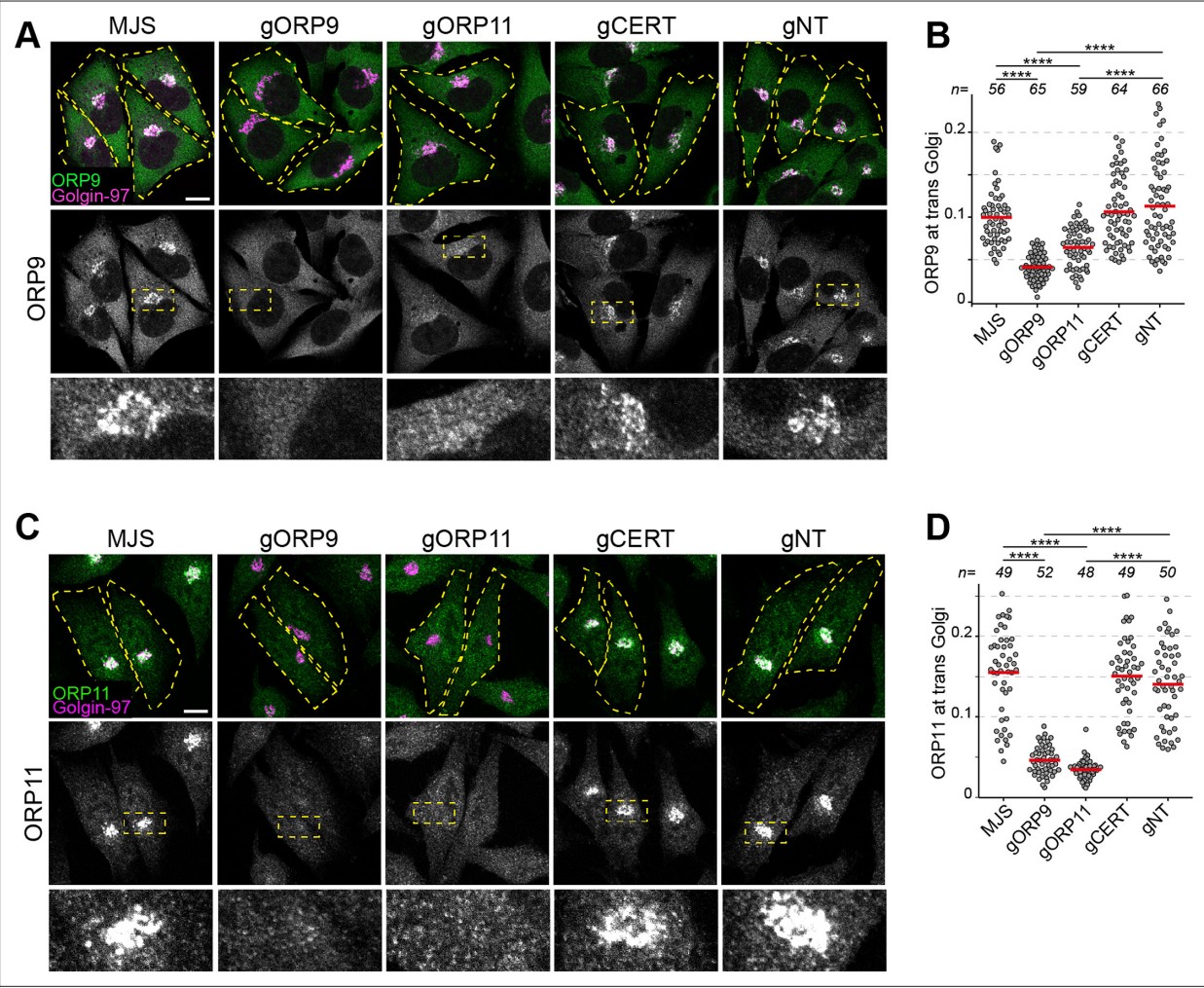

**Figure 5.** ORP9 and ORP11 dimerization is critical for their Golgi localization. (**A**) Immunofluorescence images of cells stained for ORP9. ORP9 fails to localize at the Golgi as efficiently in ORP11 knockout cells. (**B**) Colocalization analysis of immunofluorescent images corresponding to panel A. Red lines correspond to mean values from three experiments; n is the number of analyzed cells. (**C**) Immunofluorescence images of cells stained using an ORP11 antibody. Similar to ORP9, ORP11 fails to localize at the Golgi as efficiently in ORP9 knockout cells. Note that the effect of ORP9 loss on ORP11 localization is more dramatic than vice versa. (**D**) Colocalization analysis of immunofluorescent images from panel C. Red lines correspond to mean values from three experiments; n is the number of analyzed cells. All scale bars are 10 μm.

the dimerization of ORP9 and ORP11 via their coils is required for the localization of ORP9-ORP11 dimer to the ER and the *trans*-Golgi simultaneously.

## ORP9 and ORP11 are essential for PS and PI(4)P levels in the Golgi apparatus

ORP9 and ORP11 belong to the OSBP-related protein (ORP) family that transfers sterols or PS while transferring phosphatidylinositol phosphates in the opposite direction (*Wang et al., 2019*; *Mesmin et al., 2013*; *de Saint-Jean et al., 2011*; *Maeda et al., 2013*; *Figure 6A*; *Figure 6—figure supplement 1B*). ORP9 and ORP11 carry the conserved PS binding site and are recently shown to relocate to the site of lysosomal damage to supply PS for membrane repair (*Tan and Finkel, 2022*). Using an in vitro lipid transfer assay, we confirmed that the lipid transfer domains (ORD) of ORP9 and ORP11 are sufficient to traffic PS in vitro (*Figure 6B and C*). ORP9 and ORP10, but not ORP11, are shown to traffic PI(4)P (*Kawasaki et al., 2022*; *He et al., 2023*). Using a PI(4)P transfer assay that utilizes the PI(4)P-binding PH domain of FAPP1, we show that ORD of ORP11 can also traffic PI(4)P between model membranes (*Figure 6D and E*; *Figure 6—figure supplement 1C,D*; *Ikhlef et al., 2021*). Furthermore, similar to ORP5, ORP8, and ORP10, the PS-trafficking activity of ORP9- and ORP11-ORDs was improved when acceptor liposomes were decorated with PI(4)P (*Kawasaki et al., 2022*; *Chung et al., 2015*; *Moser von Filseck et al., 2015*; *Figure 6—figure supplement 1E*). This indicated that ORDs of ORP9 and ORP11 transport PI(4)P in the opposite direction of PS trafficking.

To demonstrate the contribution of ORP9-ORP11 dimerization to their lipid transfer ability at membrane contact sites, we reconstituted these sites in vitro by attaching a His-tagged VAPB devoid of its transmembrane helix to donor liposomes and providing full-length ORP9 and ORP11 as tethers to the acceptor liposomes (*Figure 6F*, *Figure 6—figure supplement 1F–H*). PS transfer assay in these reconstituted contact sites showed that as a trimer, the ORP9-ORP11-VAPB complex displays an improved lipid transfer capacity (*Figure 6G and H*).

PS is synthesized in the ER and its concentration increases along the secretory pathway as it is enriched on the cytosolic leaflet of the plasma membrane (*Holthuis and Menon, 2014*). PI(4)P, on the other hand, is abundant in the *trans*-Golgi, where it recruits many proteins to this membrane (*Posor et al., 2022*). Another LTP localized at the ER-*trans* Golgi membrane contact sites, OSBP transfers cholesterol from the ER to the *trans*-Golgi, while counter-transporting PI(4)P to the ER for its hydrolysis by the ER-resident PI(4)P phosphatase SAC1 (Figure 8). Homologous to OSBP, the ORP9-ORP11 dimer resides in the ideal intracellular interface to traffic PS in anterograde and PI(4)P in retrograde direction between the ER and *trans*-Golgi (Figure 8). This also suggested that the loss of either protein would cause PS and PI(4)P imbalances between the ER and *trans*-Golgi. We tested this notion by using PI(4)P- and PS-binding biosensors (*Yeung et al., 2008*; *Moorhead et al., 2010*; *Figure 6—figure supplement 1I,J*). Golgi localization of these sensors indicated that PS levels were decreased in the Golgi of these cells (*Figure 6I and J*). Also, PI(4)P showed accumulation in the Golgi of ORP9 and ORP11 knockout cells (*Figure 6K and L*). We confirmed the PI(4)P accumulation by quantifying the Golgi localization of a PI(4)P-specific antibody (*Figure 6—figure supplement 1J*). Interestingly, both phenotypes, lowered PS and increased PI(4)P levels, were better pronounced in ORP9 knockout cells compared to ORP11 knockouts. Furthermore, both phenotypes could be rescued by reconstitution of the missing protein (*Figure 6J and L*). Overall, these findings suggested that the ORP9-ORP11 dimer is required for maintaining PS and reducing PI(4)P levels in the Golgi apparatus.

## De novo sphingomyelin synthesis in the Golgi is impaired in ORP9 and ORP11 knockouts

Next, we set out to investigate the possible role of ORP9 and ORP11 in regulating cellular sphingomyelin levels. By performing a sphingomyelin synthase activity assay that uses the fluorescent ceramide analog NBD-ceramide as a substrate (*Kol et al., 2017*), we found that ORP9 and ORP11 knockout cells do not have reduced sphingomyelin synthesis capacity (*Figure 7A and B*). The same was also found in CERT knockout cells. Whereas all three knockouts demonstrated increased glucosylceramide synthesis capacity, supporting the lipidomics analysis (*Figure 3A*). Unaltered sphingomyelin synthesis capacity was further validated by the unreduced protein and mRNA levels of both human sphingomyelin synthases, SMS1 and SMS2 (*Figure 7—figure supplement 1A,B*).

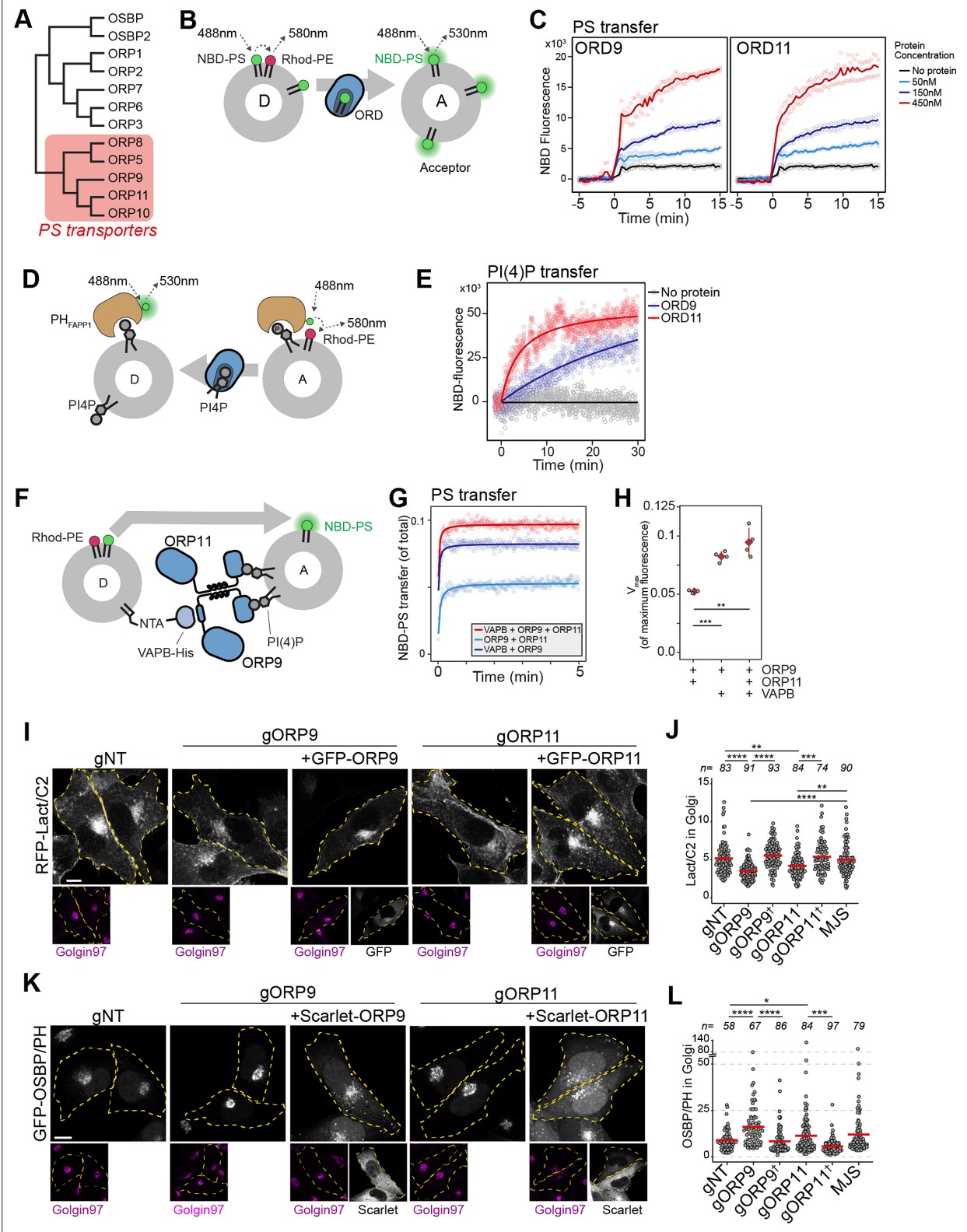

**Figure 6.** ORP9 and ORP11 are essential for phosphatidylserine (PS) and phosphatidylinositol-4-phosphate (PI(4)P) levels in the Golgi apparatus. (**A**) Phylogenetic tree of human OSBP-related domains showing that ORP9 and ORP11 belong to the PS transporter branch. (**B**) Schematic representation of FRET-based PS transfer assay. Rhodamine in the donor vesicles quenches NBD fluorescence unless NBD-labeled lipid is transferred to acceptor liposomes. (**C**) FRET-based lipid transfer assay using the OSBP-related domains of ORP9 and ORP11, ORD9, and ORD11, respectively,

*Figure 6 continued on next page*

*Figure 6 continued*

demonstrate the PS trafficking ability between vesicles in vitro. (**D**) Graphical representation of PI(4)P transfer assay that utilizes the NBD-labeled PH domain of FAPP1. PI(4)P transfer by a protein leads to increase NBD-fluorescence by dequenching. (**E**) PI(4)P transfer assay using ORP9-ORD and ORP11-ORD (ORD9 and ORD11, respectively). (**F**) Schematic representation of in vitro reconstitution of membrane contact sites for lipid transfer assays. (**G**) PS transfer assay at in vitro reconstituted membrane contact sites formed by the VAPB-ORP9-ORP11 trimer. (**H**) Vmax quantifications of PS transfer assays from panel E. (**I**) The PS sensor Lact-C2 localizes less prominently to the Golgi in ORP9 and ORP11 knockout cells. (**J**) Golgi quantification of the PS sensor RFP-LactC2 indicates reduced PS levels in this organelle. This phenotype could be rescued by reconstitution of the missing protein in knockout cells (shown with a dagger). Red lines correspond to mean values from three experiments; n is the number of analyzed cells. (**K**) The PI(4)P sensor OSBP-PH localizes more prominently to the Golgi area in ORP9 and ORP11 knockout cells. (**l**) Golgi quantification of the PI(4)P sensor confirms the accumulation of PI(4)P at this organelle. This phenotype was rescued by reconstitution of the missing protein in knockout cells (shown with a dagger). Red lines correspond to mean values from three experiments; n is the number of analyzed cells.

The online version of this article includes the following source data and figure supplement(s) for figure 6:

**Figure supplement 1.** ORP9 and ORP11 are needed for maintaining phospholipid levels in the Golgi.

**Figure supplement 1—source data 1.** Raw unedited gels for *Figure 6—figure supplement 1*.

**Figure supplement 1—source data 2.** Uncropped and labeled gels for *Figure 6—figure supplement 1*.

SMS1 is localized at the *trans*-Golgi, while SMS2 mainly localizes at the plasma membrane but is also found at the *trans*-Golgi (*Tafesse et al., 2006*). As the enzymatic activity assay in lysates reports on the global sphingomyelin synthesis capacity, we next investigated de novo sphingomyelin synthesis that occurs at the *trans*-Golgi, where CERT as well as the ORP9-ORP11 dimer localize. For this purpose, we chased the metabolic fate of palmitic acid alkyne (*Figure 7C*). Palmitic acid is the precursor of sphinganine and all sphingolipids (*Figures 3B and 7C*). To enter the sphingolipid pathway, palmitic acid first needs to travel to the ER and later to the Golgi for conversion to sphingomyelin, unlike ceramide analogs, such as NBD-ceramide, that can be converted to sphingomyelin by the plasma membrane-resident SMS2. Furthermore, the palmitic acid alkyne does not contain a bulky fluorescent group and the terminal alkyne allows visualization by click chemistry (*Figure 7C*). As this method enabled us to monitor de novo sphingomyelin production in intact cells, we confirmed the reduced de novo sphingomyelin synthesis in CERT knockout cells using this method (*Figure 7D and E*, *Figure 7—figure supplement 1C*). In addition, ORP9 and ORP11 knockouts also showed decreased conversion of palmitic acid to sphingomyelin, suggesting a similar defect as CERT knockouts. To confirm the decreased sphingomyelin synthesis, we quantified the Golgi localization of a DAG biosensor, C1ab domain of protein kinase D1, that can report on sphingomyelin synthesis (*Capasso et al., 2017*; *Peretti et al., 2008*). This showed decreased DAG levels in the Golgi that can be accounted for lowered sphingomyelin synthesis (*Figure 7—figure supplement 1D*).

The observations that loss of ORP9 or ORP11 does not affect CERT protein (*Figure 3E*) or reduce CERT localization to the Golgi (*Figure 7—figure supplement 2*) implied that CERT-mediated transfer routes are not affected in ORP9 or ORP11 knockouts. Lipidomics analysis showing accumulation of ceramide and glucosylceramide in ORP9/ORP11 knockouts further suggested the conversion of ER-bound ceramides to sphingomyelin in the Golgi is hampered.

To distinguish between two possibilities that may result in lowered sphingomyelin levels – a ceramide delivery defect to the *trans*-Golgi or an inability to convert ceramide to sphingomyelin in the Golgi, we established an immunoisolation protocol for *trans*-Golgi membranes using a monoclonal antibody against the *trans*-Golgi marker Golgin-97 (*Figure 7F*). These isolates were enriched for the *trans*-Golgi marker Golgin-97 and were devoid of other organelle markers, such as the ER, mitochondria, and lysosomes (*Figure 7G*). We confirmed the membrane integrity of Golgin-97-enriched fractions by their sphingomyelin synthase activity, as these multi-pass transmembrane proteins require intact membranes for activity (*Tafesse et al., 2006*; *Figure 7—figure supplement 3A*). Lipidomics analysis of these fractions validated that the loss of ORP11 and especially that of ORP9 results in lowered PS levels in the *trans*-Golgi (*Figure 7H*, *Figure 7—figure supplement 3B–D*). The *trans*-Golgi fractions also showed reduced sphingomyelin levels, confirming the limited de novo sphingomyelin synthesis (*Figure 7D,E,I*, *Figure 7—figure supplement 3E*). Moreover, the same fractions showed elevated ceramide levels in these fractions (*Figure 7J*, *Figure 7—figure supplement 3F*), revealing that the knockouts do not have a ceramide delivery defect, but instead, a lowered capacity to convert ceramide to sphingomyelin in their *trans*-Golgi membranes. We tested this notion by performing a sphingomyelin synthase activity assay in the isolated Golgi membranes, where supplying excess

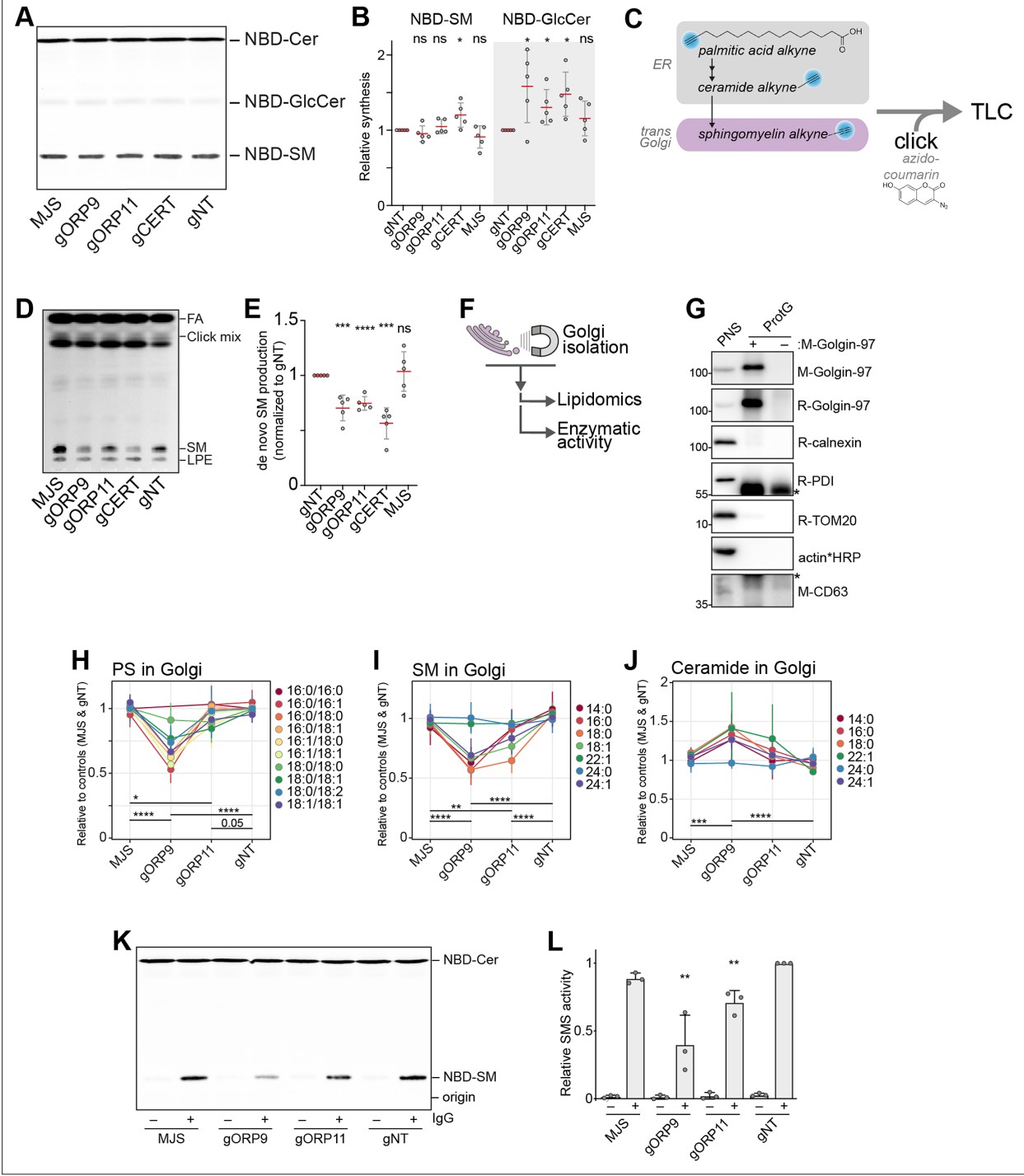

**Figure 7.** ORP9 and ORP11 are needed for de novo sphingomyelin synthesis in the Golgi apparatus. (**A**) Thin-layer chromatography readout of the enzymatic activity assay performed in lysates reveals the unreduced sphingomyelin synthesis capacity of knockout cells. (**B**) Quantification of the enzymatic activity assays corresponding to panel A. Note the increased GlcCer production capacity in the knockout cells. Red lines and error bars correspond to mean and standard deviation from five experiments, respectively. (**C**) Graphic representation of experiments using palmitic acid alkyne for monitoring de novo sphingomyelin synthesis. Alkyne-modified sphingolipids are 'clicked' with azido-coumarin before thin-layer chromatography (TLC) analysis. (**D**) TLC readout of de novo sphingomyelin synthesis assay in intact cells using palmitic acid alkyne. Knockout cells demonstrate reduced conversion of palmitic acid to sphingomyelin. FA: fatty acid, SM: sphingomyelin, LPE: lyso-O-phosphatidylethanolamine (**E**) Quantification of de novo sphingomyelin synthesis assay from panel C. Red lines and error bars correspond to mean and standard deviation from five experiments, respectively. (**F**) Simplified representation of Golgi-targeted lipidomics and enzymatic activity assays. (**G**) Western blot confirming the immunomagnetic isolation of *trans*-Golgi membranes. Cross-reactivity to Protein G and/or Golgin-97 IgG were labeled with asterisks. (**H–J**) Lipidomics analysis of Golgi isolates shows

*Figure 7 continued on next page*

Figure 7 continued

lowered PS levels in the Golgi of ORP9 and ORP11 knockout cells. Same Golgi fractions have reduced sphingomyelin but increased ceramide levels. Each colored line corresponds to a lipid species. Data points and the error bars correspond to mean and standard deviations from four experiments. (**K**) TLC readout of enzymatic activity assay performed in isolated Golgi from knockout cells. Golgi of ORP11 and especially that of ORP9 demonstrate lowered capacity to synthesize sphingomyelin. (**L**) Quantification of sphingomyelin synthesis activity assay performed in Golgi isolates corresponding to panel K. Bars and error bars denote mean and standard deviations from three experiments, respectively.

The online version of this article includes the following source data and figure supplement(s) for figure 7:

**Source data 1.** Raw unedited gels for *Figure 7*.

**Source data 2.** Uncropped and labeled gels for *Figure 7*.

**Figure supplement 1.** Loss of ORP9 or ORP11 does not affect sphingomyelin synthase levels.

**Figure supplement 1—source data 1.** Raw unedited gels for *Figure 7—figure supplement 1*.

**Figure supplement 1—source data 2.** Uncropped and labeled gels for *Figure 7—figure supplement 1*.

**Figure supplement 2.** CERT is not less present in the Golgi of ORP9 and ORP11 knockouts.

**Figure supplement 2—source data 1.** Raw unedited gels for *Figure 7—figure supplement 2*.

**Figure supplement 2—source data 2.** Uncropped and labeled gels for *Figure 7—figure supplement 2*.

**Figure supplement 3.** Immunomagnetic isolation yields to intact Golgi membranes.

**Figure supplement 4.** Golgi localization of sphingomyelin synthases is not reduced in ORP9 and ORP11 knockout cells.

substrate to these membranes bypasses the ceramide delivery routes (*Figure 7K and L*). This assay further substantiated that the Golgi of ORP11 and especially of ORP9 knockout cells have a lowered sphingomyelin synthesis capacity. Meanwhile, localization of both sphingomyelin synthases was not reduced in the Golgi, as detected by three different antibodies (*Figure 7—figure supplement 4*). Collectively, our results indicate that PS-PI(4)P exchange between the ER and *trans*-Golgi mediated by the ORP9-ORP11 LTP complex is critical for de novo sphingomyelin synthesis in trans-Golgi.

## Discussion

In this study, we describe a gene knockout library for the systematic characterization of intracellular LTPs. The arrayed design of the library enables high and low throughput analysis. As new LTPs are identified on a regular basis, the arrayed feature of the library allows expansion to include newly identified LTPs, e.g., ATG2A, SHIP164, KIAA0100 (Hobbit in *Drosophila*), KIAA1109 (Csf1 in yeast), and RMDN3 are new human LTPs that are identified through the course of this study (*Valverde et al., 2019*; *Osawa et al., 2019*; *Yeo et al., 2021*; *Castro et al., 2022*; *Neuman et al., 2022*).

Lipidomics analysis of the library demonstrated many lipid imbalances. We further validated the loss of NPC1, NPC2, CERT, and GLTP with sphingolipid imbalances (*Hanada et al., 2003*; *Nurmi et al., 2023*; *Newton et al., 2018*; *Figure 1C*). In addition to sphingolipids imbalances, we found STARD7 knockout cells with increased levels of phosphatidylglycerol (PG), a lipid class exclusive to mitochondria in mammals (*Figure 1C*). STARD7 is a phosphatidylcholine (PC) transfer protein that localizes to the mitochondrial intermembrane space to supply PC to the mitochondrial inner membrane (*Horibata and Sugimoto, 2010*; *Saita et al., 2018*; *Horibata et al., 2016*). Consequently, loss of STARD7 reduces PC levels in this membrane as well as decreases cardiolipin levels and respiratory capacity (*Saita et al., 2018*; *Horibata et al., 2016*). While PC in the inner membrane is not a precursor of cardiolipin, it is possible the reduced PC levels in this membrane impair the activity of the cardiolipin synthase that uses PG as its substrate, suggesting an explanation for the PG accumulation in STARD7 knockout cells.

LTPs often localize at membrane contact sites to facilitate lipid exchanges between organelles. We found ORP9 and ORP11 localizing at the ER-*trans* Golgi contact sites to exchange PS and PI(4)P. We show that the dimerization is critical for the contact site localization, and consequently, loss of either protein is sufficient to disturb PS and PI(4)P levels in the Golgi. In return, this phospholipid imbalance reduces sphingomyelin synthesis capacity of the Golgi. Compared to ORP11 knockout cells, ORP9 knockout cells display more pronounced phenotypes of cellular sphingolipid imbalances (sphingomyelin and glucosylceramide), phospholipid imbalances in the Golgi apparatus (PS and PI(4)P), as well as reduced capacity of sphingomyelin synthesis in trans-Golgi membranes. Furthermore, the loss of

ORP9 leads to a more dramatic effect on the Golgi localization of ORP11 than vice versa. This is most likely due to ORP9 providing a critical factor, i.e., the FFAT motif, for contact site localization, whereas it is possible the loss of ORP11 can be partially compensated by ORP10.

The contact site localization of the ORP9-ORP11 dimer is homologous to that of the ORP9-ORP10 dimer localizing at the ER-endosome and ER-Golgi contact sites asORP10 also uses its coiled coils to interact with ORP9 *Kawasaki et al., 2022*; *He et al., 2023*. In both LTP complexes, ORP9 provides the FFAT motif required for ER contact. ORP10 can also deliver PS to Golgi membranes (*Venditti et al., 2019*), however, our lipidomics analysis of ORP10 knockout cells did not demonstrate any sphingolipid imbalances (*Figure 8—figure supplement 1A*). Besides ORP9, ORP10, and ORP11, other LTPs are also shown to dimerize at the ER-Golgi interface. A recent structural study has revealed the architecture of a previously-described OSBP dimerization that is mediated by a central core domain containing two alpha helices (*Ridgway et al., 1992*; *de la Mora et al., 2021*). CERT forms a similar dimer and the conformational changes dictated by the central core domain are responsible for regulating the activity of the CERT dimer (*Gehin et al., 2023*). Both OSBP and CERT homodimers form a T-shaped structure, dissimilar to the heterodimers of ORP9, ORP10, and ORP11.

What distinguishes ORP11 from ORP10? An analysis of expression profiles from >900 cancer cell types showed that ORP10 and ORP11 are not differentially expressed (*Figure 8—figure supplement 1B*). On the contrary, they show a positive correlation. It is possible that ORP9 interacts with ORP10 or ORP11 in different biological conditions or sub-cellular locations. ER-endosome contact site localization of the ORP9-ORP10 dimer supports this notion (*Kawasaki et al., 2022*). Even within the Golgi, ORP9-ORP10, and ORP9-ORP11 dimers might be facilitating lipid flows at different sub-organellar domains. The presence of multiple phosphatidylinositol (PI) transfer proteins localized at ER-*trans*-Golgi contact sites feeding PI(4)P synthesis in the Golgi for distinct purposes and the observation of different PI(4)P-containing sub-domains in the Golgi promotes the idea that ORPs could localize at different Golgi sub-domains (*Mizuike and Hanada, 2024*; *Cockcroft and Lev, 2020*). Furthermore, all three proteins, together with OSBP and ORP1L, are recruited to the lysosomal damage site in response to the increased PI(4)P levels on lysosomal delimiting membrane (*Tan and Finkel, 2022*; *Radulovic et al., 2022*). Collectively, these reports highlight the spatiotemporally dynamic nature of PI(4)P-mediated lipid trafficking and suggest that the two dimers could have different functions.

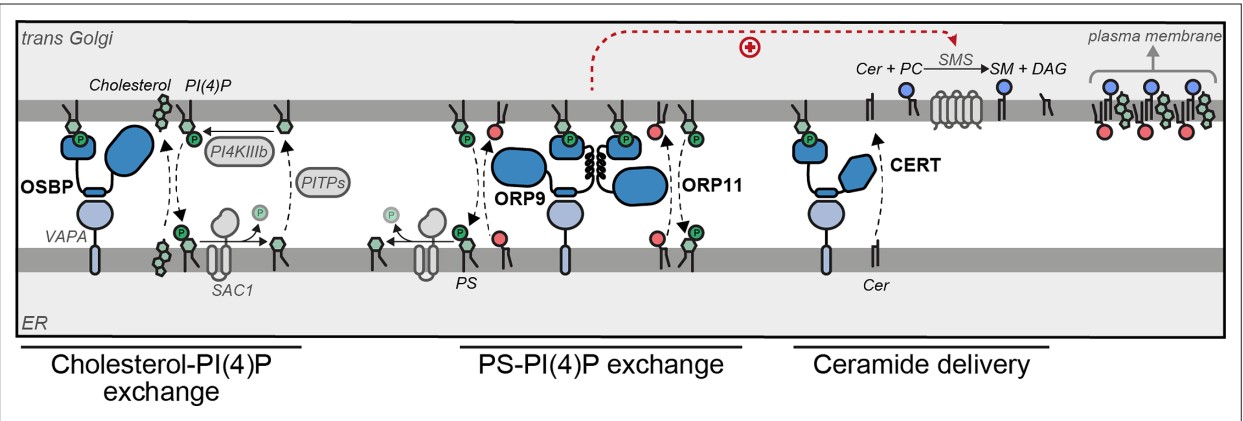

**Figure 8.** Model of ORP9-ORP11 mediated promotion of sphingomyelin synthesis at the ER-trans-Golgi contact site. ORP9 and ORP11 require dimerization with each other to localize at ER-*trans*-Golgi membrane contact sites, where they exchange phosphatidylserine (PS) for phosphatidylinositol-4-phosphate (PI(4)P). Consequently, loss of either protein causes PS and PI(4)P imbalances in the Golgi apparatus. Since only ORP9 contains a FFAT motif, its loss leads to a more pronounced effect on PS, PI(4)P, and sphingolipid levels. The same contact site accommodates other LTPs. Multiple phosphatidylinositol transfer proteins (PITPs) traffic phosphatidylinositol to the Golgi for their phosphorylation. Each PITP at the ER-Golgi contact site serves a different function—one of these proteins, Nir2, also contains a FFAT motif (*Peretti et al., 2008*; *Peretti et al., 2019*). OSBP and CERT are responsible for the anterograde trafficking of cholesterol and ceramide, the latter is used for sphingomyelin production. Cholesterol, sphingomyelin, and PS are trafficked to the plasma membrane by vesicular means to be asymmetrically distributed between the leaflets of the plasma membrane.

The online version of this article includes the following figure supplement(s) for figure 8:

**Figure supplement 1.** ORP10 knockouts do not display sphingolipid imbalances.

An intriguing observation is that the loss of ORP9 causes more accumulation of ceramide than CERT knockouts, despite the loss of either protein leading to a comparable reduction of sphingo-myelin levels. As the loss of CERT would lead to a ceramide accumulation in the ER, it is possible that such an accumulation in the ER is sensed to reduce ceramide production in this organelle. An ER-resident candidate ceramide sensor for this purpose was suggested previously (*Vacaru et al., 2009*). Meanwhile, our findings show a ceramide accumulation in the *trans*-Golgi of ORP9 knockouts. Accordingly, ceramide accumulation in the Golgi caused by the loss of ORP9 would fail to '*turn on*' an ER-localized ceramide sensing machinery, thus leading to further ceramide accumulation compared to CERT knockout cells. This notion also supports our finding that the loss of the ORP9-ORP11 dimer causes ceramide accumulation primarily in the Golgi.

Asymmetric distribution of lipids between two bilayer leaflets is a characteristic feature of the plasma membrane. This is owed to the build-up of sphingomyelin and cholesterol on the outer leaflet and PS on the inner leaflet (*Holthuis and Menon, 2014*). The transition of the thin, symmetrical ER membrane to a thicker, rigid, and asymmetrical one takes place in the Golgi (*Figure 8*; *Holthuis and Menon, 2014*). Sphingolipid, cholesterol, and PS concentrations also increase along the secre-tory pathway. Various mechanisms are described to drive the sphingolipid and cholesterol trafficking against the concentration gradient, including thermodynamic trapping of cholesterol due to complex formation with sphingomyelin or energy release from the retrograde trafficking of PI(4)P followed by its hydrolysis in the ER (*Holthuis and Menon, 2014*; *Mesmin et al., 2013*; *Mesmin et al., 2017*). The same PI(4)P gradient between the ER and Golgi could also power the anterograde trafficking of PS, which, unlike sphingomyelin or cholesterol, is still exposed on the cytosolic leaflet of the Golgi where it has a higher concentration than the ER – thus, maintaining lipid trafficking against the gradient requires energy. In brief, our finding that the ORP9-ORP11-dimer-mediated phospholipid exchange promotes the sphingomyelin synthesis reveals further intertwining of lipid gradients along the secre-tory pathway at the ER-*trans*-Golgi membrane contact sites.

## Materials and methods
### Cell culture
MelJuSo and HEK293T (ATCC CRL-3216) cells were cultured in IMDM (Gibco #21980) and DMEM (Gibco #41966) supplemented with 8% fetal calf serum (Biowest #S1810), respectively. MelJuSo cell line authentication was performed by Eurofins Genomics (19-ZE-000487).

### Library design, generation, and transduction
Targeting and non-targeting guide sequences (*Supplementary file 1*) were obtained from the Brunello Human CRISPR Knockout Pooled Library *Doench et al., 2016* and cloned into lentiCRISPR v2 plasmid (*Sanjana et al., 2014*) as described previously (*Wijdeven et al., 2022*). Cloning of guide RNAs were individually confirmed by Sanger sequencing.

For library transduction, HEK cells were seeded on a 96-well plate on the day before transfection and transfected with library plasmids and lentiviral packaging plasmids pRSVrev, pHCMV-G VSV-G, and pMDLg/pRRE using polyethyleneimine (Polysciences #23966). A day after replacing the medium, the virus was harvested and MelJuSo cells were transduced in the presence of 4 µg/ml polybrene (EMD Milipore #TR-1003-G). Transduced cells were subjected to selection using 2 µg/ml puromycin.

### Lipid extractions and lipidomics analysis
For lipidomics analysis, library-generated cells were expanded to 15 cm dishes and cultured in IMDM supplemented with lipid-depleted serum (Pel-Freez #37217–5) for 3–4 days. Next, cells were scraped in 2% NaCl solution and lipids were extracted following the Bligh-Dyer protocol. In brief, harvested cells were resuspended in 200 µL 2% (w/v) NaCl followed by the addition of 500 µL methanol and 250 µL chloroform. Samples were vortexed for 5 min. Following phase separation by adding 250 µL chloroform and 250 µL 0.45% (w/v) NaCl, samples were centrifuged for 5 min at 15,000 × g. Next, bottom fractions were collected and dried under a continuous nitrogen stream.

Comprehensive, quantitative shotgun lipidomics was carried out as described in detail elsewhere (*Ghorasaini et al., 2021*; *Su et al., 2021*). Briefly, dried lipid extracts were spiked with 54 deuterated internal standards and dissolved in methanol: chloroform 1:1 containing 10 mM ammonium acetate.

Lipids are then analyzed with a flow injection method at a flow rate of 8 µL/min applying differential ion mobility for lipid class separation and subsequent multiple reaction monitoring in positive and negative electrospray ionization mode. Using the Shotgun Lipidomics Assistant (SLA) software individual lipid concentrations are calculated after correction for their respective internal standards.

### Data analysis and statistics

Data analyses, including lipidomics data analysis, were performed using R (4.1.0 'Camp Pontanezen') and RStudio (2022.12.0+353) with the following packages: *ggplot2*, *dplyr*, *readr*, *ggrepel*, *ggcorrplot2*, *reshape2*, *ggbeeswarm*, *ggsignif*, *purr*, *tidyr*, and *tibble*. For Z-score calculations, first, the percentages of each lipid class within individual measurements were calculated, followed by calculating Z-scores for individual data points using the following formula: '$Z = (x - µ)/σ$,' where $µ$ is the mean percentage of a lipid class and $σ$ is the standard deviation. Next, average Z-scores for LTP knockout cell lines were calculated. For POPC normalized data analysis for the analysis of acyl chain distributions and Golgi lipidomics, each lipid subspecies normalized to palmitoyl-oleoyl-phosphatidylcholine (POPC) levels. For Golgi lipidomics, POPC-normalized values were normalized again to the mean of POPC-normalized values for MJS and gNT samples. Raw lipidomics data is available in *Supplementary file 2*.

Mean values are denoted in all graphs except for boxplots. When present, error bars indicate standard deviations. In boxplots, middle line denotes median, box boundaries denote the first (Q1) and third (Q3) quartiles, and the lower and upper whiskers denote '$Q1-1.5*IQR$' and '$Q3+1.5*IQR$,' respectively. IQR: inter-quartile range. For statistical analysis, student's t-test is used, unless stated otherwise. $*p<0.05$, $**p<0.01$, $***p<0.001$, $****p<0.0001$.

### Co-immunoprecipitation, SDS-PAGE, and western blotting

Co-immunoprecipitation from HEK293T cells, SDS-PAGE, and western blotting were performed as described previously (*Cabukusta et al., 2020*).

### Immunofluorescence staining and quantifications

For equinatoxin staining, cells were fixed with 4%PFA: PBS (v/v) and permeabilized with 10 µg/ml digitonin in PBS for 15 min at RT. Further stainings were performed in PBS. PI4P staining was performed after fixation in 2% PFA/PBS for 15 min followed by permeabilization with 20 µM digitonin in Buffer A (20 mM PIPES, pH 6.8, 137 mM NaCl, and 2.7 mM KCl). Blocking was performed in Buffer A supplemented with 5% (v/v) serum. Next, cells were incubated with the PI(4)P antibody and the anti-mouse IgM secondary antibody, respectively–both diluted in a blocking solution. Cells were washed with buffer A and fixed in 2% PFA/PBS for 5 min. Coverslips were mounted on slides using Vectashield Vibrance (Vector Laboratories #H-1700). All other immunofluorescence staining was performed as described previously (*Cabukusta et al., 2020*).

Images were acquired using a Zeiss LSM 900 with Airyscan. Images were analyzed and quantified using ImageJ/FIJI software. To quantify the Golgi localization of lipid biosensors (OSBP-PH, LactC2, and C1ab), first, a mask was created using Golgin-97 counter staining and this mask was used to quantify signal from *trans*-Golgi as well as the cytoplasm. Golgi localization of immunofluorescence staining of endogenous proteins (ORP9, ORP11, CERT) and the PI(4)P antibody was performed first by creating a mask using Golgin-97 and later using this mask to quantify the signal of interest relative to the signal of Golgin-97. Manders' coefficient for colocalization was calculated using the Jacop plugin for ImageJ/FIJI.

### Flow cytometry analysis

MelJuSo cells were brought to suspension by trypsinization and stained with the indicated probes on ice prior to analysis using a BD LSR-II equipped with 488 and 561 nm lasers. Data was analyzed using FlowJo v.10 software.

### Electron microscopy

Cells cultured in 6 cm dishes were fixed for an hour at room temperature by adding double-concentrated fixative to the medium (final concentration: 1.5% glutaraldehyde in 0.1 M cacodylate buffer). After three times rinsing with 0.1 M cacodylate buffer, the cells were postfixed with 1% osmium tetroxide

and 1.5% uranyl acetate. Cells are dehydrated with a series of ethanol, followed by a series of mixtures of ethanol and EPON (LX112, Leadd), and at the end pure EPON. BEEM capsules filed with EPON were placed on the dishes with the open face down. After EPON polymerization at 40 °C the first night and 70 °C the second night, the BEEM capsules were snapped off. Ultrathin sections 80 nm were made parallel to the surface of the BEEM capsules containing the cultured cells. The sections were contrasted with uranyl acetate and lead hydroxide and examined with a Tecnai Twin transmission electron microscope (Thermo Fisher, Eindhoven, Netherlands). Overlapping images were automatically collected and stitched together into a composite image as previously described (*Faas et al., 2012*).

## Recombinant protein expression and purification

A codon-optimized version of cDNA encoding the non-toxic version of equinatoxin (*Deng et al., 2016*) was ordered from IDT technologies, a GFP sequence was obtained from the meGFP-C1 vector and a cDNA encoding the fusion protein was synthesized using Gibson Assembly. A codon-optimized version of cDNA encoding the PH domain of FAPP1 containing the T13C, C37S, and C94Smutations were ordered from IDT technologies. cDNA encoding the human VAPB gene was described previously (*Cabukusta et al., 2020*). cDNAs encoding ORD domains of ORP9 and ORP11, VAPB without its C-terminal transmembrane helix, FAPP1-PH, and EQT-GFP fusion protein were cloned into a pNKI1.8/GST-expression vector.

For FAPP1-PH and VAPB-His expressions *E. coli* BL21 was grown in LB medium and expression was induced when $OD_{600}$ was at 0.6 for overnight at 18 °C. For other proteins, *E. coli* BL21/Rosetta were grown in 2x YT medium, and expression was induced when $OD_{600}$ was at 1 for overnight at 18 °C. Bacterial pellets were resuspended in GST purification buffer (50 mM Tris pH 8.0, 250 mM NaCl, 1 mM EDTA, 1 mM DTT) and lysed by tip sonication. Lysates were cleared by centrifugation at 12,000 × g for 30 min. Proteins were purified using Glutathione Sepharose 4 Fast Flow (GE Healthcare #17-5132-03) and cleaved using 3 C protease protein prior to reverse purification using glutathione beads followed by a HiLoad 16/60 Superdex 75 size exclusion chromatography.

Full-length ORP9 and ORP11 were expressed using a baculovirus expression in *Spodoptera frugiperda* (Sf9) using an adapted Bac-to-Bac system (Invitrogen). In brief, cDNAs were cloned to an in-house insect expression vector and bacmids were generated using EmBacY cells (Geneva Biotech) isolated using isopropanol precipitation. 10 μg Bcmid was transfected to sedentary Sf9 cells using CellFectin (Invitrogen) in SFM-II medium (Gibco) in a 6-well plate at 28 °C. Virus-containing medium was harvested (P0) after 72 hr for infection of $10^6$ cells in Insect-Express medium (Lonza) to be cultured at 28 °C while constant shaking. Cells were harvested by centrifugation after 72 hr and purified using Strep-Tactin Sepharose beads (iba #2-1201-010).

## NBD-labeling of FAPP1-PH

Labeling of FAPP1-PH was performed according to *Ikhlef et al., 2021*. In brief, purified protein was incubated with fivefold excess IANBD-amide overnight (Setareh Biotech #6281) followed by quenching of unbound dye with *L*-cysteine fivefold excess of the dye. Next, the labeled protein was subjected to the Zeba Spin desalting column (Thermo Fisher #89890). Labeling efficiency was determined by absorbances at 280 nm and 480 nm using a NanoDrop.

## Methyl-β-cyclodextrin and MTT viability assay

Cells were seeded in a 96-well plate and were cultured in OptiMEM (Thermo Fisher #31985047) for 3 days. Cells were treated with 10 mM methyl-β-cyclodextrin (Sigma #C4555) and cell viability was tested using MTT viability assay (Cayman #21795). Absorbance at 560 nm was measured using a BMG ClarioStar plate reader.

## Liposome preparations

For PS transfer assay using ORDs, donor liposomes composed of 2% NBD-PS (Avanti #810194), 2% Rhodamine-PE (Avanti #810150), 10% DOPE (Avanti #850725), 86% DOPC (Avanti #850375), and acceptor liposomes composed of DOPC with or without 5% brain PI(4)P (Avanti #840045) were used. For PS transfer assays using full-length proteins, donor liposomes composed of 2% NBD-PS, 2% Rhodamine-PE, 2% 18:1 DGS-NTA(Ni) (Avanti# 790404), 10% POPE (Avanti # 850757) and 84% POPC (Avanti #850457) and acceptor liposomes of 85% POPC, 10% POPE, and 5% brain PI4P were used.

For PI4P transfer assays, donor liposomes were 95% DOPC and 5% POPS (Avanti # 840034) and acceptor liposomes were 93% DOPC and 5% PI4P (Echelon #P-4016), and 2% Rhodamine-PE.

Lipids were mixed in a glass container and dried under a constant nitrogen flow to create a film. Lipids were freeze-dried at least for 2 hr under a vacuum. Next, lipid films were rehydrated in HKM (50 mM HEPES pH 7.2; 120 mM potassium acetate; 1 mM MgCl$_2$) buffer for 30 min followed by five cycles of freeze-thaw. Lipids were extruded using Avanti MiniExtruder using a 100 nm or 200 nm filter. Uniformity of lipids were confirmed using a Wyatt Nanostar Dynamic Light Scattering.

## Lipid transfer assays

For PS transfer assays using ORDs, final concentrations of 80 µM of each donor and acceptor vesicles were used in 100 µL volume and the indicated amounts of protein of interest was added in 5 µL. NBD fluorescence in time was measured using a BMG ClarioStar plate reader. For PS transfer assay using full-length proteins, 100 µM of each donor and acceptor vesicles were used with 150 nM ORP9, 150 nM ORP11, and 450 nM VAPB-His. Maximum rate of transfer (Vmax) was calculated using a self-starting non-linear least squares function implemented in R (SSmicmen) after background subtraction to no protein control and normalizing to a maximum amount of fluorescence obtained after addition of 2% CHAPS. PI4P transfer assay were adapted from *Ikhlef et al., 2021*. Donor and acceptor liposomes were first incubated with 500 nM NBD-labeled FAPP1-PH for 5 min before addition of 150 nM ORD9 or ORD11. Data was fit using a self-starting non-linear least squares function implemented in R (SSmicmen).

## Liposome floatation

10 µM VAPB-His was incubated 360 µM liposomes containing NTA-DGS lipids in 100 µL HKM buffer for 30 min at RT before mixing with 0.3 mL HKM buffer containing 54% sucrose to obtain a final concentration of 40.5% sucrose. Samples were laid at the bottom of an ultracentrifuge tube and cushioned with HKM buffer containing 30% and 0% sucrose. Samples were centrifuged at 200,000 × *g* for 2 hr at 4 °C and fractions were collected for Coomassie blue staining.

## Liposome tethering

100 nM of each ORP9, ORP11, and VAPB-His were incubated with donor and acceptor liposomes for 1 hr at RT. Diameter of liposomes were measured using a Wyatt Nanostar Dynamic Light Scattering.

## AlphaFold and AlphaFold-multimer predictions

Structure predictions were performed using AlphaFold, AlphaFold-Multimer, and ColabFold prediction tools, as described (*Jumper et al., 2021*; *Mirdita et al., 2022*; *Evans et al., 2021*).

## Golgi isolation and quantification of Golgi lipid levels

Cells were washed twice with 0.25 M Sucrose and scraped in IM buffer (0.25 M Mannitol, 0.5 mM EGTA, 5 mM HEPES pH 7.4). Samples were passed through a 26x G needle for lysis. Lysates were cleared from nuclei by centrifugation twice at 600 × g for 5 min. Post-nuclear supernatant was subjected to protein determination using BCA protein determination kit. Equal amounts of protein were supplemented with 150 mM NaCl and incubated with Protein A/G Magnetic Beads (Thermo Fisher #88802) preloaded either or not with Golgin-97 antibody. Next, samples were washed three times with IM buffer supplemented with 150 mM NaCl before lipid extraction for lipidomics analysis (described above) or enzymatic activity assay (described below). Amounts of Golgi fractions isolated from different knockouts were confirmed by western blotting.

## siRNA transfections and metabolic chasing of BODIPY-C5-ceramide in intact cells

siRNA transfections were carried out as described previously *Cabukusta et al., 2020*. siRNA for UGCG (M-006441-02-0005) and UGT8 (M-010270-02-0005) were obtained from Horizon Discovery. MelJuSo cells treated with siRNAs or 10 µM PDMP (Cayman #62595) were washed twice with PBS and incubated with serum-free medium supplemented with 0.25 µM BODIPY-C5-Ceramide complexed to BSA (Thermo Fisher #B22650). After 3 hr, cells were subjected to lipid extraction and thin-layer chromatography as described below. The following lipids were used as standards:

TopFlour-C11-galactosylceramide (Avanti Polar Lipids #810266), TopFlour-C11-glucosylceramide (Avanti Polar Lipid #810267), and BODIPY- C5-Lactosylceramide (Thermo Fisher #B34402).

## Metabolic chasing of de novo sphingomyelin synthesis

Cells were incubated with 20 μM ethanolic palmitic acid alkyne (Cayman Chemical #13266) in serum-free medium for 6 hr. Next, cells were washed twice in PBS and scraped in 2%NaCl solution. Lysates were subjected to protein determination using Pierce BCA Protein Kit (Thermo #23225) and equal amounts of proteins were used for lipid extractions as above. For alkaline hydrolysis, dried lipids were resuspended in 200 μL methanolic sodium methoxide and incubated at room temperature for 1 hr. Samples were added 30 μL acetic acid/water (1:4, v/v), 120 μL 2% NaCl, and 400 μL chloroform to re-extract lipids. Dried lipids were resuspended in 15 μL chloroform and 65 μL of click mix containing 400 μM 3-Azido-7-hydroxycoumarin (Baseclick #BCFA-047–1) and 900 μM tetrakis(acetonitrile)copper(I) tetrafluoroborate (Sigma #677892) in acetonitrile/ethanol (7:3, v/v) was added and incubated 3 hr at 45 °C prior to TLC analysis.

## Quantitative RT-PCR

RT-PCR was performed as described previously *Cabukusta et al., 2020* using the following primer sets:

> CERT_1: ATGTCGGATAATCAGAGCTGGA / ATCCTGCCACCCATGAATGTA,
> CERT_2: TCCATCTGTCTTAGCAAGGCT / GCTGTTCAATGGCATCTATCCA,
> SMS1_1: TGTGCCGAGTCTCCTCTGA / CCGTTCTTGTGTGCTTCCAAA,
> SMS1_2: CAGCATCAAGATTAAACCCAACG / TGGTGAGAACGAAACAGGAAAG,
> SMS2_1: TCCTACGAACACTTATGCAAGAC / CCGGGTACTTTTTGGTGCCT,
> SMS2_2: CAAATTGCTATGCCCACTGAATC / GTTGTCAAGACGAGGTTGAAAAC

## Enzymatic activity assay in lysates and isolated Golgi fractions

For enzymatic activity measurements in lysates, cells were washed twice in 0.25 M Sucrose and scraped in IM buffer (0.25 M Mannitol, 0.5 mM EGTA, 5 mM HEPES pH 7.4). Samples were passed through a 26x G needle for lysis. Lysates were cleared from nuclei by centrifugation twice at 600 × g for 5 min. Post-nuclear supernatant was subjected to protein determination using BCA protein determination kit. Equal amounts of protein in a 50 μL volume were mixed with 50 μL IM buffer supplemented with 5 μM NBD-C6-ceramide (Avanti Polar Lipids #8102109) from an ethanolic solution. Reactions were incubated for 1 hr at 37 °C in dark with constant shaking. Next, samples were subjected to lipid extractions and thin-layer chromatography as described. For the enzymatic activity assays in Golgi, isolated Golgi fractions were resuspended in 50 μL IM buffer and incubated with NBD-C6-ceramide as above.

## Thin-layer chromatography

Dried lipids were spotted on a thin-layer chromatography (TLC) Silica gel 60 (Merck #1.05554.0001) plate and developed in chloroform:methanol:acetone:acetic acid:water (50:10:20:10:5, v/v/v/v/v). Fluorescent images were acquired using a Typhoon FLA9500 equipped with a 488 nm laser and a BPB1 filter (530DF20).

## Gene expression analysis

Gene expression profiles from 947 cancer cell lines were obtained from the Cancer Cell Line Encyclopedia (*Barretina et al., 2012*).

## Plasmids

| Plasmid | Reference | cDNA | Backbone |
| --- | --- | --- | --- |
| GST-EQT-GFP | This study | IDT GeneBlock | pNKI1.8 |

*Continued on next page*

*Continued*

| Plasmid | Reference | cDNA | Backbone |
|---|---|---|---|
| Scarlet-ORP9 iso.2 | This study | Mouse Osbpl9 isoform 2 cDNA was ordered from Horizon Discovery. BC023759. IMAGE:5344071 | mScarlet-C1 |
| Scarlet-ORP11 | | Human OSBPL11 cDNA was ordered from Horizon Discovery. IMAGE:3916115. BC065213 | mScarlet-C1 |
| GFP-OSBP-PH | addgene #49571, *Moorhead et al., 2010* | | |
| RFP-LactC2 | addgene#74061, *Yeung et al., 2008* | | |
| mCherry-PKD/C1ab | addgene# 139314, *Zewe et al., 2018* | | |
| GFP-CC11 | This study | | |
| Mito-tagRFP | This study | cDNA encoding the first 70 amino acids of TOM70 was ordered from IDT technologies. | tagRFP-N1 |
| Mito-CC9-tagRFP | This study | | tagRFP-N1 |
| GFP-PH9 | This study | | meGFP-C1 |
| GFP-PH11 | This study | | meGFP-C1 |
| GFP-PH9-PH11 | This study | | meGFP-C1 |
| FLAG-ORP11 | This study | | FLAG-C1 |
| GST-ORD9 | This study | | pNKI1.8 |
| GST-ORD11 | This study | | pNKI1.8 |
| GST-FAPP1-PH | This study | IDT GeneBlock | pNKI1.8 |
| GST-VAPBΔTM-His | This study | *Cabukusta et al., 2020* | pNKI1.8 |
| Strep-ORP9 | This study | | |
| Strep-ORP11 | This study | | |
| FLAG-VAPA | *Cabukusta et al., 2020* | | |
| meGFP-C1 | *Cabukusta et al., 2020* | | |
| mScarlet-C1 | *Cabukusta et al., 2020* | | |
| GFP-VAPA | *Cabukusta et al., 2020* | | |
| GFP-VAPA** | *Cabukusta et al., 2020* | | |
| Scarlet-CERT | *Cabukusta et al., 2020* | | |

## Antibodies

| Target | Supplier | Catalog # | Host |
|---|---|---|---|
| OSBP | Proteintech | 11096–1-AP | Rabbit |
| ORP1L | PMID: 12631712 | | Rabbit |
| CERT | Bethyl | A300-669A | Rabbit |
| NPC1 | Novus | NB400-148 | Rabbit |

*Continued on next page*

*Continued*

| Target | Supplier | Catalog # | Host |
|---|---|---|---|
| NPC2 | Proteintech | 19888–1-AP | Rabbit |
| GLTP | Proteintech | 10850–1-AP | Rabbit |
| PITPNB | Proteintech | 13110–1-AP | Rabbit |
| ORP9 | Proteintech | 11879–1-AP | Rabbit |
| STARD7 | Proteintech | 15689–1-AP | Rabbit |
| SEC14L1 | Proteintech | 25541–1-AP | Rabbit |
| PITNM1 | Proteintech | 26983–1-AP | Rabbit |
| SEC14L2 | Origene | TA503723 | Mouse |
| TEX2 | Bethyl | A304-705A | Rabbit |
| FAPP2 | Proteintech | 15410–1-AP | Rabbit |
| ORP9 | Santa Cruz | sc-398961 | Mouse |
| CD63-V450 | BD Horizon | 561984 | Mouse |
| CTB-Alexa568 | Invitrogen | C34777 | |
| Actin | Sigma-Aldrich | A5441 | Mouse |
| GM130 | BD Transduction | 610823 | Mouse |
| FLAG-HRP | Sigma-Aldrich | A8592 | Mouse |
| GFP | *Cabukusta et al., 2020* | | Rabbit |
| Scarlet | ChromoTek | 6G6-100 | Mouse |
| tagRFP | Evrogen | AB232 | Rabbit |
| Golgin-97 | Invitrogen | A-21270 | Mouse |
| Golgin-97 | Cell Signaling | 13192 S | Rabbit |
| Calnexin (C5C9) | Cell Signaling | 2679 | Rabbit |
| R-PDI | Cell Signaling | 3501 | Rabbit |
| TOM20 | Proteintech | 11802–1-AP | Rabbit |
| Actin*HRP | Santa Cruz | sc-47778 | Mouse |
| SMS1 | Sigma/Aldrich | HPA045191 | Rabbit |
| SMS2 | Santa Cruz | sc-293384 | Mouse |
| CERT | Proteintech | 15191–1-AP | Rabbit |
| SMS1 | Proteintech | 19050–1-AP | Rabbit |
| ORP11 | Proteintech | 11318–1-AP | Rabbit |
| Anti-mouse Alexa 488 | Thermo Fisher | A21202 | Donkey |
| Anti-mouse Alexa 568 | Thermo Fisher | A10037 | Donkey |
| Anti-mouse Alexa 647 | Thermo Fisher | A21236 | Goat |
| Anti-rabbit Alexa 488 | Thermo Fisher | A21206 | Donkey |
| Anti-rabbit Alexa 568 | Thermo Fisher | A10042 | Donkey |
| Anti-rabbit Alexa 647 | Thermo Fisher | A31573 | Donkey |
| Anti-Mouse IgG, HRP | Thermo Fisher | G21040 | Goat |
| Anti-Rabbit, IgG, HRP | Thermo Fisher | G21234 | Goat |
| Anti-Mouse IgM, Alexa Fluor 488 | Thermo Fisher | A-21042 | Goat |

| Target | Supplier | Catalog # | Host |
|---|---|---|---|
| Anti-PtdIns(4)P | Echelon | Z-P004 | Mouse IgM |

## Acknowledgements

Authors thank Ruud Wijdeven, Robbert Q Kim of the LUMC protein facility, Carolina Jost, and the members of the Neefjes lab for their technical and critical input. Authors are especially grateful to Joost Holthuis and Matthijs Kol of the University of Osnabrück for their intellectual input. This research is supported by the ONCODE Institute and the Spinoza award of the Dutch Research Council (NWO 00897590) (JN).

## Additional information

### Funding

| Funder | Grant reference number | Author |
|---|---|---|
| Oncode Institute | | Jacques Neefjes |
| Nederlandse Organisatie voor Wetenschappelijk Onderzoek | Spinoza 00897590 | Jacques Neefjes |

The funders had no role in study design, data collection and interpretation, or the decision to submit the work for publication.

### Author contributions

Birol Cabukusta, Conceptualization, Data curation, Software, Formal analysis, Supervision, Investigation, Methodology, Writing – original draft, Project administration, Writing - review and editing; Shalom Borst Pauwels, Data curation, Formal analysis, Investigation, Visualization, Writing – original draft, Writing - review and editing; Jimmy JLL Akkermans, Methodology; Niek Blomberg, Aat A Mulder, Data curation; Roman I Koning, Data curation, Visualization, Methodology, Writing – original draft; Martin Giera, Conceptualization, Resources, Data curation, Investigation, Methodology, Writing – original draft, Writing - review and editing; Jacques Neefjes, Conceptualization, Resources, Funding acquisition, Investigation, Methodology, Writing – original draft, Writing - review and editing

### Author ORCIDs

Birol Cabukusta  https://orcid.org/0000-0002-1187-3048
Shalom Borst Pauwels  https://orcid.org/0000-0001-5596-2512
Aat A Mulder  https://orcid.org/0000-0002-4779-099X
Roman I Koning  https://orcid.org/0000-0001-6736-7147
Martin Giera  https://orcid.org/0000-0003-1684-1894
Jacques Neefjes  https://orcid.org/0000-0001-6763-2211

Reviewer #1 (Public Review): https://doi.org/10.7554/eLife.91345.3.sa1
Reviewer #2 (Public Review): https://doi.org/10.7554/eLife.91345.3.sa2
Author response https://doi.org/10.7554/eLife.91345.3.sa3

## Additional files

### Supplementary files

• Supplementary file 1. Overview of lipid transfer proteins (LTPs) targeted in this library and the list of short guide RNA sequences used in the knockout library to target LTPs.

• Supplementary file 2. Dataset file for the lipidomics analysis of lipid transfer protein (LTP) knockout cell lines.

• MDAR checklist

## Data availability

Guide sequences used to target lipid transfer proteins and the lipidomics data generated in this study is included in the supplementary files.

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
