## [Editor Report · eLife assessment]

This **valuable** manuscript systematically addresses the role of intracellular lipid transfer proteins on cellular lipid levels. It provides **convincing** evidence on the role of ORP9 and ORP11 in sphingolipid metabolism at the Golgi complex. This article will be of broad interest to cell biologists interested in lipid metabolism and membrane biology.

---

## [Referee Report · Reviewer #1 (Public Review)]

Summary:

In this well-designed study, the authors of the manuscript have analyzed the impact of individually silencing 90 lipid transfer proteins on the overall lipid composition of a specific cell type. They confirmed some of the evidence obtained by their own and other research groups in the past, and additionally, they identified an unreported role for ORP9-ORP11 in sphingomyelin production at the trans-Golgi. As they delved into the nature of this effect, the authors discovered that ORP9 and ORP11 form a dimer through a helical region positioned between their PH and ORD domains.

Strengths:

This well-designed study presents compelling new evidence regarding the role of lipid transfer proteins in controlling lipid metabolism. The discovery of ORP9 and ORP11's involvement in sphingolipid metabolism invites further investigation into the impact of the membrane environment on sphingomyelin synthase activity.

Weaknesses:

There are a couple of weaknesses evident in this manuscript. Firstly, there's a lack of mechanistic understanding regarding the regulatory role of ORP9-11 in sphingomyelin synthase activity. Secondly, the broader role of hetero-dimerization of LTPs at ER-Golgi membrane contact sites is not thoroughly addressed. The emerging theme of LTP dimerization through coiled domains has been reported for proteins such as CERT, OSBP, ORP9, and ORP10. However, the specific ways in which these LTPs hetero and/or homo-dimerize and how this impacts lipid fluxes at ER-Golgi membrane contact sites remain to be fully understood.

Regardless of the unresolved points mentioned above, this manuscript presents a valuable conceptual advancement in the study of the impact of lipid transfer on overall lipid metabolism. Moreover, it encourages further exploration of the interplay among LTP actions across various cellular organelles.

---

## [Referee Report · Reviewer #2 (Public Review)]

Summary:

The authors set out to determine which lipid transfer proteins impact the lipids of Golgi apparatus, and they identified a reasonable number of "hits" where the lack of one lipid transfer protein affected a particular Golgi lipid or class of lipids. They then carried out something close to a "proof of concept" for one lipid (sphingomyelin) and two closely related lipid transfer proteins (ORP9/ORP11). They looked into that example in great detail and found a previous unknown relationship between the level of phosphatidylserine in the Golgi (presumably trans-Golgi, trans-Golgi Network) and function of the sphingomyelin synthase enzyme. This was all convincingly done - results support their conclusions - showing that the authors achieved their aims.

Impact:

There are likely to be 2 types of impact:

(I) cell biology: sphoingomyelin synthase, ORP9/11 will be studied in future in more informed ways to understand (a) the role of different Golgi lipids - this work opens that out and produces a to more questions than answers (b) the role of different ORPs: what distinguishes ORP11 from its paralogy ORP10?

(ii) molecular biochemistry: combining knockdown miniscreen with organelle lipidomics must be time-consuming, but here it is shown to be quite a powerful way to discover new aspects of lipid-based regulation of protein function. This will be useful to others as an example, and if this kind of workflow could be automated, then the possible power of the method could be widely applied.

Strengths:

Nicely controlled data;

Wide-ranging lipidomics dataset with repeats and SDs - all data easily viewed.

Simple take home message that PS traffic to the TGN by ORP9/11 is required for some aspect of SMS1 function.

Weaknesses:

Model and Discussion:

Despite the authors saying that this has been addressed in their rebuttal, I still struggle to find any ideas about the aspect of SMS1 function that is being affected.

As I mentioned before, even if no further experiments were carried out the authors could discuss possibilities. one might speculate what the PS is being used for. For example, is it a co-factor for integral membrane proteins, such as flippases? Is it a co-factor for peripheral membrane proteins, such as yet more LTPs? The model could include the work of Peretti et al (2008), which linked Nir2 activity exchanging PI:PA (Yadav et al, 2015) to the eventual function of CERT. Could the PS have a role in removing/reducing DAG produced by CERT?

---

## [Author Response]

The following is the authors’ response to the original reviews.

**Reviewer #1 (Recommendations For The Authors):**
The authors should possibly discuss more the other cases when LTPs of the same type of ORP9 and ORP10 have been found to dimerise. They should definitely cite and discuss the evidence reported in February this year in CMLS (see https://link.springer.com/article/10.1007/s00018-023-04728-5). In this paper, authors reported very similar findings as those the authors have in Figures 3, 4, S6, S7, and S8. Specifically, in this CMLS paper the authors find that ORP9 and ORP10 (not ORP11) interact through a central helical region and that ORP9 localises ORP10 to the ER-Golgi MCSs by providing ORP10 with a binding site for VAPs, where the heterodimer mediates the exchange of PtdIns(4)P for PtdSer.

We thank the reviewer for their recommendations. The mentioned paper has simply gone unnoticed by us and is now referred in the revised manuscript. Various other papers reporting on LTP dimerizations are already cited in our manuscript: ORP9-ORP10 dimerization (Kawasaki et al. 2022), ORP9-ORP11 dimerization (Zhou et al. 2010), and ORP9-ORP10/11 dimerization (Tan and Finkel 2022). Revised manuscript now discusses the dimerization of CERT and OSBP while citing Gehin et al. 2023, Ridgway et al. 1992 and de la Mora et al. 2021.

**Reviewer #2 (Recommendations For The Authors):**
Model and Discussion:Give an idea about the aspect of SMS1 function that is being affected. Even if no further experiments were carried out, the authors could discuss possibilities. One might speculate what the PS is being used for. For example, is it a co-factor for integral membrane proteins, such as flippases? Is it a co-factor for peripheral membrane proteins, such as yet more LTPs? The model could include the work of Peretti et al (2008), which linked Nir2 activity exchanging PI:PA (Yadav et al, 2015) to the eventual function of CERT. Could the PS have a role in removing/reducing DAG produced by CERT?

We thank the reviewer for their recommendations. The same recommendations were also scripted in the public review, which we believe we answered sufficiently.

Other, Minor:Make clear that there is no sterol readout (Fig 1C)

We would like to point out that Figure 1C has a sterol readout as CE refers to cholesterol esters.

PH domains of ORP9 and ORP11 localized only partially to the Golgi, unlike the PH domains of OSBP and CERT" (line 154). Say here where the non-Golgi ORP9 and ORP11 PH domain pool is - presumably in the cytoplasm.

We thank the reviewer for their suggestion and rephrase the sentence accordingly.

Fig 7H-J: histograms not lines as these are separate unlinked categories

We thank the reviewer for their suggestion. However, we think the original figure represent our findings in the best possible way. Our analysis regarding individual lipid species is also included in Supplementary figure 10.

**Reviewer #3 (Recommendations For The Authors):**
(1) At the end of the intro, in summarizing their findings, the authors state (p3. lines 48-49) "These findings highlight how phospholipid and sphingolipid gradients along the secretory pathway are linked at ER-Golgi membrane contact sites." This should instead read "These findings highlight THAT phospholipid and sphingolipid gradients along the secretory pathway are linked at ER-Golgi membrane contact sites."

We thank the reviewer for their suggestion and change the sentence accordingly.

(2) As noted in the public section, to show that ORP9/11 do indeed exchange lipids, an in vitro experiment demonstrating that ORP11 can transfer PI4P is essential. Ideally, it would be best to examine PS AND PI4P transfer by ORP9 AND 11 separately AND then by the ORP9/11 heterodimer. This could lend insights as to the function of the heterodimer. The He et al et Yu paper should provide guidelines for this. Why have the heterodimers?

We believe we addressed this point by showing the lipid transfer ability of the ORP9-ORP11 dimer. These findings are now part of the revised manuscript.

(3) It would be interesting to discuss the roles of ORP9/ORP11 versus ORP9/ORP10... they seem so analogous, although this is at the discretion of the authors.

We thank the reviewer for their suggestion. Since the difference between ORP9-ORP10 and ORP9-ORP11 dimers was also raised by other reviewers, we decided to include this discussion in the manuscript. A section based on our answer to Reviewer #2 in Public Review is now part of the Discussions.

(4) The authors used a melanoma cell line in their screens (p3, line 59). Could they explain why they used this cell line versus others?

We chose MelJuSo cell for various reasons. Mainly, MelJuSo are diploid, which eases generating knockouts in a screening setup compared to other polyploid cancer cell lines (e.g. HeLa). Furthermore, our CRISPR/Cas9 screening protocols are optimized for these cell lines.